# Parallel evolution in the emergence of highly pathogenic avian influenza A viruses

Marina Escalera-Zamudio [1,4 ✉], Michael Golden [1,4], Bernardo Gutiérrez [1], Julien Thézé[1], Jeremy Russell Keown [2], Loic Carrique [2], Thomas A. Bowden[2] & Oliver G. Pybus [1,3 ✉]

Parallel molecular evolution and adaptation are important phenomena commonly observed in viruses. Here, we exploit parallel molecular evolution to understand virulence evolution in avian influenza viruses (AIV). Highly-pathogenic AIVs evolve independently from low-pathogenic ancestors via acquisition of polybasic cleavage sites. Why some AIV lineages but not others evolve in this way is unknown. We hypothesise that the parallel emergence of highly-pathogenic AIV may be facilitated by permissive or compensatory mutations occurring across the viral genome. We combine phylogenetic, statistical and structural approaches to discover parallel mutations in AIV genomes associated with the highly-pathogenic phenotype. Parallel mutations were screened using a statistical test of mutation-phenotype association and further evaluated in the contexts of positive selection and protein structure. Our resulting mutational panel may help to reveal new links between virulence evolution and other traits, and raises the possibility of predicting aspects of AIV evolution.

---

[1] Department of Zoology, Oxford University, Parks Rd, Oxford OX1 3PS, UK. [2] Division of Structural Biology, Wellcome Centre for Human Genetics, Oxford OX3 7BN, UK. [3] Department of Pathobiology and Population Sciences, Royal Veterinary College, London, UK. [4]These authors contributed equally: Marina Escalera-Zamudio, Michael Golden. ✉email: marina.escalerazamudio@zoo.ox.ac.uk; oliver.pybus@zoo.ox.ac.uk

Avian influenza A viruses (AIVs) infect bird populations worldwide[1]. While most AIVs cause only a mild-to-asymptomatic infection, highly pathogenic (HP) avian influenza, restricted to the H5 and H7 AIV subtypes, causes extremely high mortality (up to 100%) in domestic bird populations[1]. Both low-pathogenic (LP) and HP viruses can cause severe disease in humans following avian-to-human transmission[2,3]. For example, the emergence of H7N9 AIV in China caused >1500 infections and 615 human deaths between 2013 and early 2018 [4]. Low virulence AIVs have evolved into HP strains on multiple occasions, resulting in geographically- and temporally distinct outbreaks that cause substantial economic losses to the poultry industry, and pose a risk of avian-to-human spread[5,6]. It is therefore important to understand the mechanisms driving the evolution of virulence in AIVs.

Virulence in influenza A viruses is, in general, a complex trait that involves interactions among multiple viral proteins[7–9]. However, the AIV HP phenotype is an unusually simple and well-characterised molecular marker of virulence, and is therefore an attractive system in which to study virulence evolution[10]. The HP phenotype is conferred by a polybasic cleavage site (pCS) within the viral haemagglutinin (HA) protein. The HA cleavage site is located in the C-terminal domain of HA (Arg-343 in H1 numbering) and is highly conserved among all influenza A viruses, including AIVs[11]. The insertion of one or more basic amino acids at this site creates a pCS, which renders the HA protein a target for broadly expressed host proteases, facilitating systemic virus spread within avian hosts[12]. These insertions, and the subsequent emergence of a pCS, have occurred several times during the evolutionary history of AIVs: 25 different pCS sequences have been described for HP H5NX viruses since 1961, and 24 pCS have been described for HP H7NX viruses since 1963 [11]. However, the evolutionary pathways that lead to the independent, parallel evolution of a pCS, and thus the HP phenotype, are not well understood.

Parallel evolution describes repeated evolutionary changes leading to the same phenotype or genotype in independent populations, and can result from adaptation by natural selection to similar selective pressures or ecological niches[13,14]. We use the term 'parallel molecular evolution' to refer to the convergent or parallel molecular genetic changes that lead to the emergence of same phenotype[15]. RNA viruses exhibit high rates of mutation (i.e. the rate at which errors are made during the virus genome replication), large population sizes, high replication rates, consistently varying environments, and strong selective pressures. These factors, combined with small genome sizes, may restrict the number of mutational pathways that can lead to a given genotype, and may result in parallel molecular evolution being common in RNA viruses[16].

We hypothesise that the parallel evolution of the HP genotype from LP ancestors in AIV subtypes H5 and H7 may have a genetic basis, and may be therefore associated with permissive or compensatory secondary mutations occurring elsewhere in the AIV genome. Such mutations could arise prior to, or immediately after, the emergence of a pCS, perhaps to compensate for mutations with antagonistic pleiotropy, or which have detrimental effects on protein structure or folding stability[17,18]. Little is known about the genetic predisposition of AIV to acquire a pCS (except for the role of RNA structure at the cleavage site region itself[19]), nor about the role of mutations elsewhere in the virus genome that might favour the evolution of the HP phenotype.

The main goal of this study is to use a combination of analytical methods, from evolutionary and structural biology, in order to detect parallel mutations across the AIV genome that are positively associated with the evolution of the HP phenotype (hereafter termed HP-cluster associated parallel mutations, or HAPMs). We develop, validate and implement a phylogenetic model of binary trait evolution that tests whether genotype-to-phenotype associations observed across multiple lineages are statistically significant. This approach explicitly incorporates the phylogenetic correlation among observations and thus is less susceptible to error than standard statistical tests[20]. We further investigate if the HAPMs that we identify had evolved under positive selection, and use existing structural data and computational modelling to explore their possible effects on protein function and/or on structure stability (Fig. 1). We identify numerous HAPMs present in HP AIV strains belonging both to the H5 and H7 AIV subtypes, many of which appear to be associated with other important viral processes, including antigenic escape and replication efficiency. We discuss how the mutational panel presented here can provide a starting point for future molecular studies for the evolution of the HP phenotype.

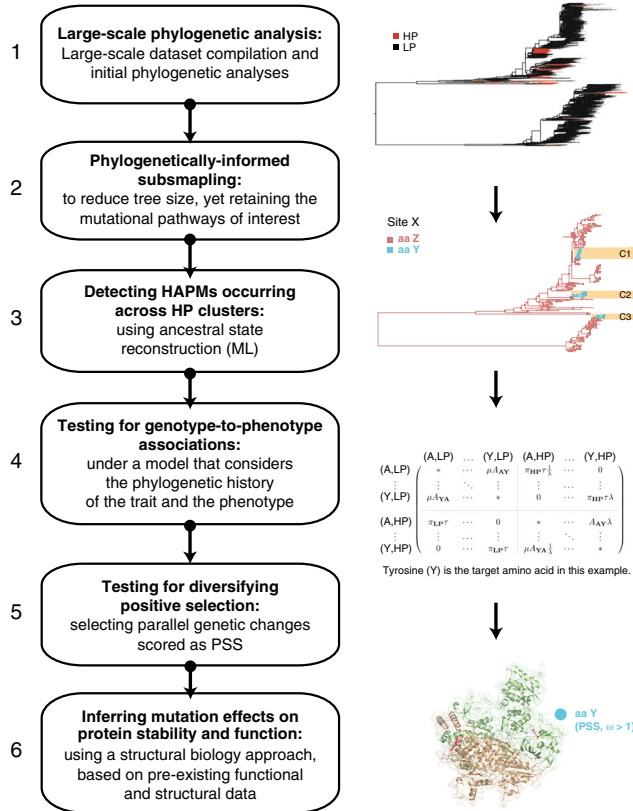

**Fig. 1 Summary of analytical approach.** A summary of our approach to identifying HAPMs (HP-cluster associated parallel mutations) in AIV genomes. 1. Large-scale data curation and phylogenetic analysis to detect genotypes and phenotypes (HP lineages highlighted in red). 2. Phylogenetically informed subsampling to reduce alignment sizes, whilst retaining HP clusters and key mutational pathways along internal branches. 3. Detecting candidate HAPMs across multiple HP clusters (highlighted in orange) through the reconstruction of ancestral states. 4. Selecting HAPMs by statistical testing for genotype-to-phenotype associations using an evolutionary-informed model (represented in the matrix by aa Y occurring in Site X). 5. Identifying HAPMs evolving under positive selection using different dN/dS ($\omega$) estimation methods. 6. Inferring the possible effects of HAPMs on protein structure stability and function, using existing structural data and experimental literature.

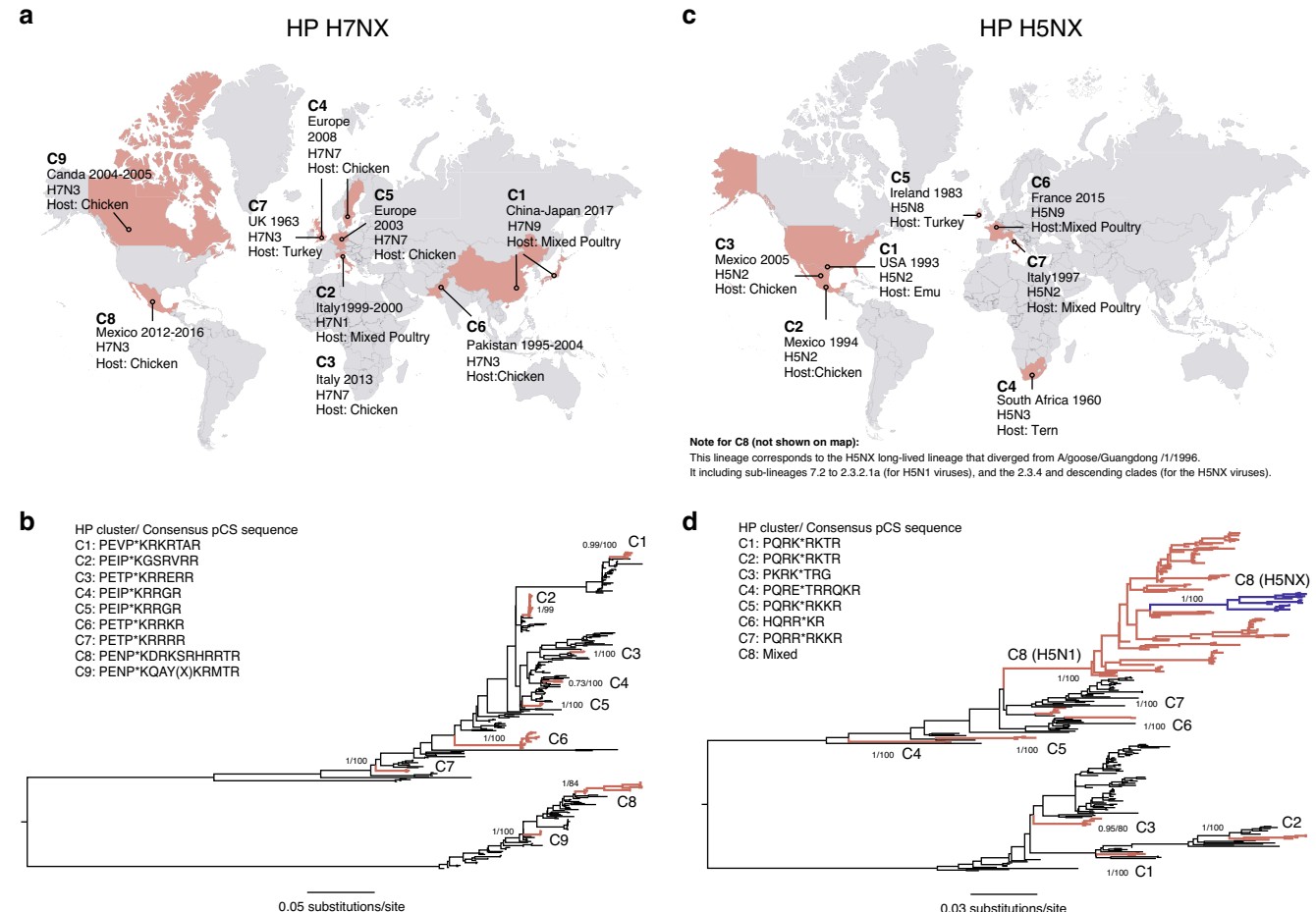

**Fig. 2 Historical occurrence of documented HP AIV outbreaks, and independent emergence of HP lineages. a** For each H7NX HP case, the recorded date of emergence and duration (in years), the country of emergence (highlighted in red), the virus subtype causing the outbreak, and the reported avian host species are indicated in the map. **b** Phylogenetic distribution of HP lineages in the H7 subsampled phylogeny. Branches leading to LP sequences are shown in black. HP sequences (highlighted in red) correspond to unique genotypes defined by a pCS (shown next to the phylogeny), as they form independent well-supported lineages (as defined in 'Methods', indicated by C1-C9, showing posterior probability and bootstrap support values >80 or 0.8 for nodes of interest), matching the historical occurrence of the documented HP outbreaks. **c** Historical occurrence of documented H5NX HP outbreaks, following the description used for panel (**a**). Data for C8 is not included, as since 1996, this lineage circulates worldwide. **d** Phylogenetic distribution of HP lineages in the H5 subsampled phylogeny, following the description used for panel (**b**). Distinct, well-supported HP clusters are highlighted in red (C1-C8). The H5NX HP viruses within cluster C8 (clade 2.3.4 and descending lineages) are highlighted in blue. Maps for the global distribution for the H5NX and H7NX HPAIV were generated with the tool http://gamapserver.who.int/mapLibrary/default.aspx under the CC BY-NC-SA 3.0 IGO license with preprint permission authorisation 362444 for WHO copyrighted material. Maps were further edited using Adobe Illustrator for the purposes needed.

## Results

**Large-scale phylogenetic analysis of independent HP clusters.** Large-scale phylogenetic analysis confirmed that H7NX viruses are split into two major clades, the Eurasian and North American lineages (Supplementary Data 1, Supplementary Fig. 1 and Fig. 2)[21]. H7 HP outbreaks occurred independently through time and space and correspond to distinct lineages that, in most cases, display a unique pCS motif (Fig. 2b)[5]. LP H7NX viruses carry a consensus amino acid sequence -PEXPKGRGLF- at the HA cleavage site, whereas a minimal consensus of -PEXPKXnK/RnRGLF- is identified for HP viruses[11]. Nine H7NX HP well-supported clusters that are spatio-temporally distinct were selected for further analyses (Fig. 2a, b). The large-scale phylogeny of H5NX viruses shows a basal clade of H5NX viruses circulating in wild birds worldwide from the late 1960s to the early 2000s (Supplementary Data 1, Supplementary Fig. 1 and Fig. 2). This basal clade is ancestral to the established HP H5N1 lineage (named C8 here), that diverged from A/goose/Guangdong /1/1996 (including sub-lineages 7.2 to 2.3.2.1a)[22].

China, this lineage has subsequently spread to Asia, Africa, Europe and America and persists until today[22]. Within C8, there is a distinct group of HP H5NX viruses, forming clade 2.3.4 and its descendants, that circulate in both wild and domestic birds[22]. Eight well-supported H5NX HP clusters that are spatio-temporally distinct were selected (Fig. 2c, d). Most HP clusters displayed a unique pCS sequence, except for C8, reflecting the ongoing evolution of this unusually long-lived lineage[22] (Fig. 2d).

Although reassortment is common among AIVs[23], our results do not provide evidence for reassortment within HP lineages themselves, as HP clusters were observed to be monophyletic and well supported in all genome segment trees (Supplementary Figs. 2–4). There are two exceptions to this; the H7N9 cluster C1 that splits into two clusters (C1.1. and C1.2) in all internal genome segment trees, and H5N1 cluster C8, which splits into two clusters (C8.1. and C8.2) in the PB2, PA and NP trees (Supplementary Figs. 2 and 3). However, due to reassortment, the internal gene segments of HP cluster strains may originate from AIV subtypes other than H7NX or H5NX. Nonetheless, low

phylogenetic support in the internal segment trees reflects that the origin of these segments cannot, in many instances, be determined (Supplementary Figs. 1 and 2). We conclude that most HP clusters for both H5NX and H7NX are short-lived, emerging and dying out within a time-span too short to allow for reassortment. An exception to this is H5NX C8 (i.e. cluster 2.3.4.4), which is a long-established HP lineage in domestic bird populations and is known to have reassorted with other AIVs[22,24], as is consequently split within internal genome segment trees.

**Identifying HAPMs significantly associated with the HP phenotype.** The number of amino acid sites that were polymorphic (i.e. present at >1% frequency) varied among virus subtypes and genome segments. For H7NX viruses, the number of polymorphic sites per segment ranged from 21 (for M1) to 125 (for HA). Maximum likelihood ancestral state reconstructions revealed 16 convergent genetic changes (i.e. candidate HAPMs) that were present in two or more H7NX HP clusters (Supplementary Data 2). For H5NX viruses, between 12 (for M1) and 141 (for HA) polymorphic sites were detected per segment. We identified 39 convergent genetic changes (candidate HAPMs) that were present in two or more H5NX HP lineages (Supplementary Data 2). Of these, 17 occurred within sub-lineages of cluster C8, and were not considered further.

To identify those HAPMs significantly associated with the HP phenotype, we developed a phylogenetic model to test the null hypothesis of no association between mutations at each position and the HP phenotype (Supplementary Text 5). For H7NX viruses, we found a strong association with the HP phenotype for 7 out of 16 mutations (see $\lambda$ and significance in Table 1). For H5NX viruses, a strong association with the HP phenotype was supported for 10 out of 22 mutations (Table 1). For some H7NX HP clusters, we identified a 20-residue deletion within the stalk region of the N1, N3 and N9 strains (Supplementary Fig. 4, Supplementary Data 2). However, this deletion was not scored as

an HAPM under our model. The frequency of stalk-deletion mutations is known to fluctuate in time and space among both LP and HP viruses[25]. Although some deletions have been experimentally linked to increased virulence, their occurrence is likely associated with host switching (from wild bird to gallinaceous hosts), rather than with the emergence of the HP phenotype[25,26].

Our ancestral state reconstructions show that HAPMs occur on the ancestral nodes of HP clusters, or on branches within HP clusters, but are not observed on LP branches that are ancestral to HP clusters (Supplementary Data 2, Supplementary Figs. 5–8 and Fig. 3). Therefore, given the phylogenetic resolution of our datasets, it is not possible to determine if HAPMs appear shortly before, during, or after the emergence of the HP phenotype. This limits our ability to determine if they are permissive or compensatory changes. The likely cause of this is a sampling bias towards HP isolates collected during outbreaks, resulting in an under-representation of the immediate LP ancestors of HP clusters. This conclusion is supported by the long branches leading to the emergence of some HP clusters, e.g. C6 H7NX and C3 H5NX (Fig. 2, Supplementary Figs. 2 and 3). In general, the ML and the MCC phylogenies were consistent.

Figure 3 illustrates the evolutionary dynamics of four selected HAPMs significantly associated with the HP phenotype. We observed high posterior probability values for ancestral amino acids at nodes preceding the emergence of HP clusters. By way of illustration, extensive parallel evolution is exhibited by mutation 143T in H7, which occurs in five HP clusters (Fig. 3a). Evolution at site 154 in H5 occurs within three HP clusters and exhibits a more complex, step-wise evolutionary process, from N to I/L to Q, reflecting a transition from LP to HP (Fig. 3b). N is present only in LP viruses, I/L occurs in HP viruses circulating prior to and during the early expansion of C8, and Q is only present in more recent HP viruses from C8. Examples of HAPMs occurring within internal genome segments are site 355 in H7NX PB2 and site 627 in H5NX PB2. Mutation 355K is present in HP clusters C5 and C6, but not in lineages representing the LP background (Fig. 3c). A similar pattern is seen for mutation 627K in the PB2

**Table 1 HAPMs significantly associated with the HP phenotype of H5 and H7 virus subtypes.**

| HAPM | Segment | HP clusters | Num. HP clusters | H7 numbering | References[a] | $\lambda$ [95% CI] | P value | Significance[b] |
|---|---|---|---|---|---|---|---|---|
| H7NX | | | | | | | | |
| R355K[c] | PB2 | C5 and C6 | 2 | | 58 | 11.0 [6.8−16.7] | <0.0001 | ** |
| A143T[c] | HA | C2, C5, C6, C7 and C8 | 5 | 125 | 38, 68−70 | 7.1 [3.3, 13.2] | <0.0001 | ** |
| V480I | PB2 | C5 and C6 | 2 | | | 8.5 [5.2−13.2] | <0.0001 | ** |
| M/L274I | HA | C4 and C8 | 2 | 256 | | 6.2 [3.1, 11.5] | 0.0024 | ** |
| S152L | PB1 | C6 and C9 | 2 | | | 5.4 [3.4−8.2] | 0.0045 | ** |
| V438I | HA | C3 and C8 | 2 | 423 | | 4.6 [2.4, 8.3] | 0.0093 | * |
| D384N | HA | C6 and C8 | 2 | 369 | | 5.1 [2.6, 9.5] | 0.018 | * |
| H5NX | | | | | | | | |
| V113I | PB1 | C2, C4 and C8 | 3 | | | 2.6 [1.6−4.5] | 0.0022 | ** |
| K379R | HA | C1 and C8 | 2 | 363 | | 14.8 [7.4, 27.5] | 0.0040 | ** |
| M242I | HA | C1 and C8 | 2 | 226 | 73 | 7.4 [3.7, 13.7] | 0.0048 | ** |
| S145L/del[d] | HA | C5 and C8 | 2 | 129 | 38, 63 | See footnote | 0.0059 | * |
| A674T | PB2 | C2 and C6 | 2 | | 66 | 5.8 [3.6−9.1] | 0.0060 | * |
| N154I/L/Q[c, e] | HA | C6, C7 and C8 | 3 | 138 | 60 | See footnote | 0.022 | * |
| F127L[c] | HA | C3, C7 and C8 | 3 | 111 | 60 | 6.5 [3.3, 12] | 0.025 | * |
| I167T | HA | C6 and C8 | 2 | 151 | 62 | 5.8 [3.0, 10.4] | 0.026 | * |
| T204I | HA | C2 and C8 | 2 | 188 | | 2.7 [1.4, 4.7] | 0.028 | * |
| E627K | PB2 | C1 and C8 | 2 | | | 2.9 [1.8−4.6] | 0.038 | * |

[a]Where such HAPMs had been previously mentioned.
[b]'*' indicates a p value < 0.05, whilst '**' indicates a p value < 0.01.
[c]Scored as positively selected site under different methods (see Supplementary Table 1 in the Supplementary Information).
[d]Individual lambda values for each site are as follows: for 145 L 77.5 [69.5, 100.0] with a p value of 0.00, and for 145del 7.0 [0.5, 14.7] with a p value of 0.04.
[e]Individual lambda values for each site are as follows: for 154I 1.7 [1.0, 2.7], for 154 L 3.2 [1.9, 5.2] and for 154Q 3.4 [2.0, 5.8], all with a p value of 0.00.

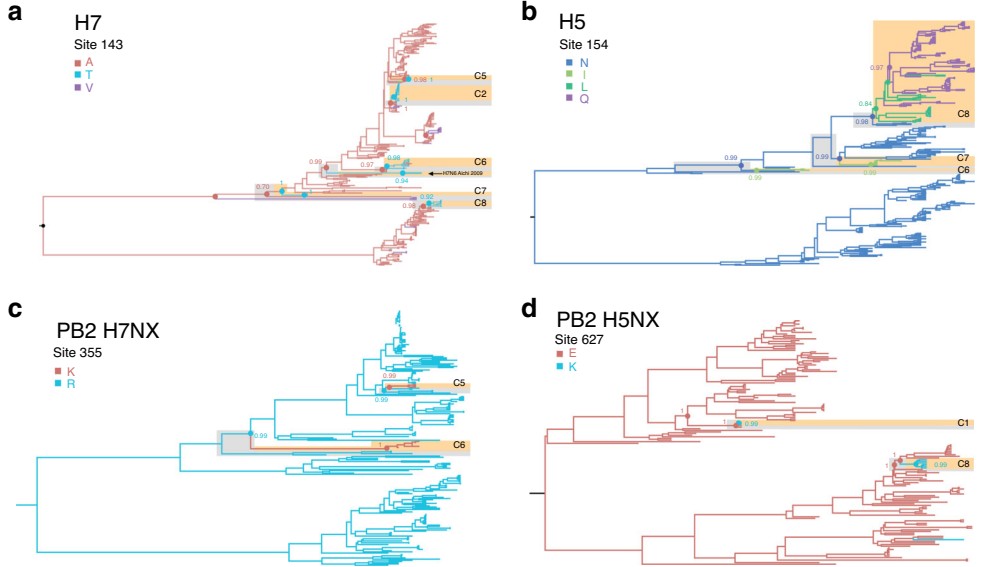

**Fig. 3 Reconstruction of amino acid evolution at selected HAPMs.** Maximum clade credibility (MCC) trees showing the reconstruction of ancestral states for four illustrative HAPMs (**a**−**d**). Branches are coloured according to the inferred ancestral amino acid states (tip states not shown). Nodes representing the immediate LP ancestors and nodes from which the HP clusters directly descend are shown with circles. The posterior probabilities for a given amino acid state occurring at the specified node are indicated. The HP sequences (HP clusters) are highlighted in orange, whilst the immediate ancestral LP sequences are highlighted in grey. For details on sampling date, location, and host species, see Supplementary Figs. 5–8.

of H5NX viruses, present in C1 and C8, but now fixed among LP virus populations (Fig. 3d). For details on sampling date, location, and host species, see Supplementary Figs. 5–8.

We further investigated if HAPMs that are significantly associated with the HP phenotype are also present in HP sequences that were not used in the subsampled alignments (Supplementary Data 3). We found that, in addition to the selected HP, mutation 154L in H5 was also fixed in two independent HP H5N1 outbreaks in Vietnam from 2008 to 2013[27], and in HP H5N2 viruses in China circulating from 2009 to 2015[28] (Supplementary Data 3). We also observed that mutation 143T is present in H7N6 viruses isolated from an outbreak in Aichi prefecture, Japan, in 2009[29] (Fig. 3a; Supplementary Data 1 and 3). Although this was not an HP outbreak, these viruses contained an additional arginine residue within the pCS (-PKGR to -PKRR), which triggered concerns about whether this lineage might evolve the HP phenotype. Fortunately, this lineage was successfully eradicated before any HP phenotype strains were observed[29]. The presence of HAPMs in these HP sequences constitute an independent test of our statistical approach.

**HAPMs associated with diversifying positive selection.** As parallel evolution is often a consequence of positive selection,[14] we investigated whether some HAPMs were inferred as being positively selected sites (PSS; see 'Methods', subsection 'Testing for diversifying positive selection'). Molecular evolutionary theory posits that most mutations within a coding sequence are expected to be deleterious, and are thus eliminated by purifying selection[30]. A smaller proportion of mutations are expected to be selectively neutral or nearly neutral, or beneficial, in which case they may be retained by directional or diversifying positive selection[30]. As expected for AIVs, and for RNA viruses in general[16], dN/dS ($\omega$) estimates for whole genome segments were <1 in all cases and the overall proportion of PSS detected was low (Supplementary Table 1 and Supplementary Data 4).

For all alignments, and under all methods, most codons exhibited $\omega \ll 1$ (strong purifying selection) or $\omega < 1$ (weaker

purifying selection or nearly neutral change). HA was the segment with the highest proportion of sites with $\omega > 1$, but even in HA most codons had $\omega \leq 1$ (86% of codons for H5 and 92% of codons for H7),[31]. FUBAR revealed 458 codons evolving under episodic negative/purifying selection in H5, compared to five sites evolving under episodic positive/diversifying selection (posterior probability of ≥0.9). For H7, 417 sites were inferred to evolve under episodic negative/purifying selection, whilst zero sites were found to be under episodic positive/diversifying selection (posterior probability of ≥0.9).

Analyses under branch-site models showed evidence for diversifying positive selection acting upon some HP clusters (H7NX clusters C4 and C7, and H5NX clusters C1, C2 and C4; Supplementary Table 1). Analyses under site models revealed for the H7NX subtype, 2/7 of the HAPMs significantly associated with the HP phenotype were identified as PSS. For the H5NX subtype, 2/10 of HAPMs were identified as PSS (see Supplementary Table 1 for details). At first impression, this suggests that most HAPMs significantly associated with the HP phenotype are not a consequence of positive selection[14,17]. However, some HAPMs may be selectively neutral in the absence of a pCS (LP phenotype), but offer a positive effect associated with a pCS (HP phenotype). Thus, they may not be inferred as a PSS using dN/dS methods, as these are rather conservative in detecting sites under positive selection in some circumstances[32]. For example, mutation E627K in PB2, which was not inferred to be a PSS, is known to have a positive effect on the adaptation of AIV to mammalian hosts and increases replication efficacy[33]. The role of selection on whole HP lineages is less clear: for example, experiments in turkey populations suggest that the HP and LP phenotypes of H7N1 viruses have indistinguishable transmission rates and basic reproductive numbers ($R_0$)[34].

**Inferring possible effects of HAPMs on protein structure and function.** Despite experimental validations, we investigated HAPMS in the context of existing virus protein structures and the experimental literature on the potential functional significance of influenza A virus mutations. Most HAPMs had negative ddG

values[35], and thus predicted to represent fairly stabilising/neutral changes to protein structure and folding (Figs. 4 and 5). For the HA of the H7NX subtype (Fig. 4a), A143T in HA is a non-conservative amino acid change located in a solvent-accessible region within the 130-loop proximal to the receptor surface. This is within antigenic pocket A contributing to the electrostatic environment needed for ligand binding. Five antigenic regions (A−E) have been identified within antigenic pocket A for the A (H1N1) pdm09 virus, with site 143 located within five residues N-terminal to antigenic site A[36]. Although it has no direct inter-action with glycans[37], mutation A143T induces glycosylation of residue 141 by adding an N-linked glycosylation motif. This N-linked residue produces steric masking of the antigenic site and may affect HA folding and stability[38]. Other HAMPs with potential functional relevance for the H7NX subtype include M/L271I in HA, located within the receptor binding subdomain[39,40], and R355K in PB2 (Fig. 4b), which is a conservative amino acid change within a solvent-exposed loop of the electrostatic surface of the cap binding domain. This site is in direct contact with first phosphate after the cap[41] and correlates with high pathogenicity of AIVs in mouse models[42].

For the HA of the H5NX subtype (Fig. 5a), HAMPs of potential functional relevance include mutation F127L in HA, which is a conservative amino acid change within the globular head of HA1 close to antigenic site 3[43]. Similarly, mutation N154I/L/Q is a non-conservative change within antigenic site 2 and involves the loss of the asparagine-linked glycosylation site[44]. Both of these correspond to variable sites related to escape mutants[43]. However, they are not crucial for the protective efficacy of H5 vaccines, suggesting that the loss of the saccharide linkage may have little impact on antigenic variation[40,44]. I167T in HA is located in the surface-exposed region, and has been related to increased infectivity in the human lower respiratory tract in HP H5N1 viruses from Egypt[45]. S145L in HA is also located in the surface-exposed region and has been linked to increased virulence in poultry[46]. This site also undergoes occasional deletion (observed in ~10% of sequences) and it has been argued to be critical for the antigenic evolution of long-lived HPAIV clusters[38]. For the PB2 protein of H5NX viruses (Fig. 5b), the HAPM E627K is located within the surface-exposed region in in the '627 domain', and is essential for viral RNA replication and transcription in a cellular context[47]. This mutation is a well-characterised host adaptation marker associated with enhanced transmission in mammalian hosts[48]; it is rarely observed among H5NX AIVs but appears frequently among zoonotic viruses that caused human infections[7]. E627K in H5NX PB2 is predicted to be destabilising for protein structure. However, it has been shown that other mutations in PB2, such as A674T (also scored as a HAPM), appear to improve the genetic stability of E627K[49]. Site 674 is located at the interface of the polymerase complex trimer and is predicted to increase strength of the PB2 protein structure[48,49].

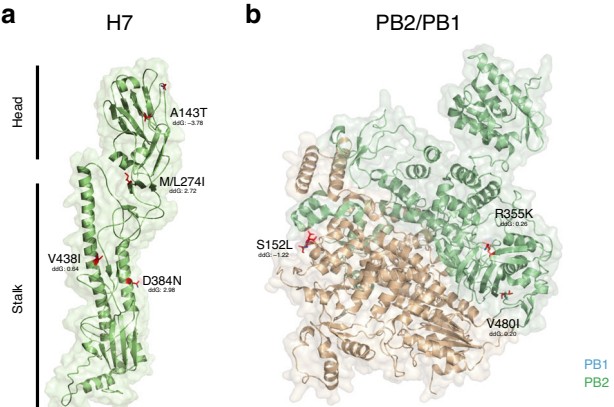

**Fig. 4 Functional relevance of HAPMs of H7NX viruses.** Protein structure models for the **a** HA (Hemagglutinin) and the **b** PB2/PB1 (Polymerase complex) of the H7NX viruses. Functionally relevant HAPMs are highlighted in red within the structural models. Only those HAMPs that are significantly associated with the HP phenotype are indicated. Estimated values for free energy change (ddG) are indicated for these sites. Negative values or those close to zero can be interpreted as fairly stabilising/neutral mutations with respect to protein structure.

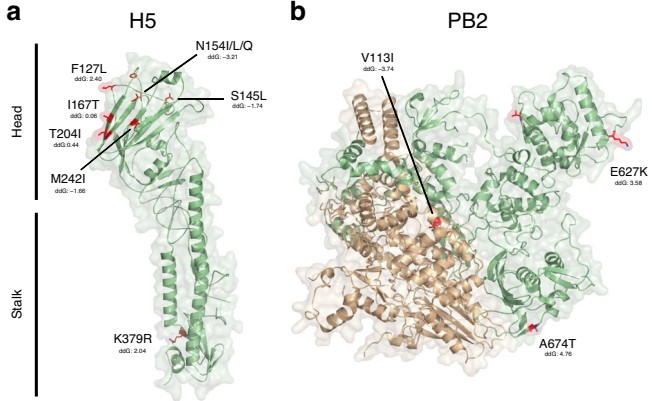

**Fig. 5 Functional relevance of HAPMs of H5NX viruses.** Protein structure models for the **a** HA (Hemagglutinin) and the **b** PB2 (Polymerase complex) of the H5NX viruses. See Fig. 4 legend for further details.

## Discussion

The occurrence of HAPMs in AIV genomes can be used as a starting point for the identification of circulating variants associated with the HP phenotype. Here we report combined evolutionary and structural evidence to support the hypothesis that some HAPMs are significantly associated with the HP phenotype, and may facilitate the evolution of HP viruses. At present we can only hypothesise on the mechanism underlying these associations and whether they might have arisen due to epistasis, which is thought to be common in RNA viruses due to their limited genome sizes[14]. As an example of within-gene epistasis, oseltamivir resistance in human H1N1 influenza A viruses is conferred by mutation H274Y in N1, yet appears to have spread only under the genetic background of two per-missive mutations (V234M and R222Q also in N1) that enable the virus to tolerate the detrimental effects of H274Y on protein structure stability[18]. Further, PB2 mutations E627K and A674T both scored as HAPMs and appear to epistatically interact: A674T counteracts the structural stability induced by E627K, whilst in vivo mouse models have shown that this mutation is a prerequisite for the acquisition of E627K, by stabilising poly-merase activity[49]. Epistatic interactions among IAV genes are much less studied but have been proposed for genes of the IAV polymerase complex[49,50].

Previous studies have noted parallel genetic changes in inde-pendent AIV lineages[51], but these were not linked to the emer-gence of the HP phenotype. One specific instance of parallel genetic change associated with the transition from an LP to an HP phenotype, giving rise to a single H7N1 HP outbreak in Italy, has been described[52–54]. We provide the first systematic and statistically rigorous screening for mutations in AIV genomes associated with HP outbreaks. We combine phylogenetically informed hypothesis testing with comparative evolutionary and

structural approaches to identify a robust set of important mutations that could be experimentally evaluated in future analyses.

Our statistical model for mutation−phenotype association reformulates the test of Bhattacharya et al.[20] as an evolutionary-informed phylogenetic model that measures the strength of association between traits. Our test uses a probabilistic approach to account for the potential founder effects (e.g. hitchhiking events) that arise from shared common ancestry, whilst maximising the statistical power to detect associations. Our model shares similarities with some discrete trait evolution models[55,56]; however, these only consider two binary traits, whereas our test associates a binary trait (HP/LP) with multiple states (20 amino acids). Thus, our model captures the potential dependence between a trait and a target amino acid using a single parameter, rather than requiring a large number of parameters to represent each possible transition separately. This simplicity results in easily interpretable hypothesis testing, straightforward inference using maximum likelihood, and lower estimation variance. Available genomes for AIV are unlikely to be randomly sampled. The LP ancestors of many HP clusters will be under sampled, and some HP lineages may not be sampled at all. However, due to the phylogenetic nature of our tests, these sampling biases will lower our statistical power to detect HAPMs, but they will not introduce false positives. Therefore, our statistical approach is conservative (some associations will not be detected), but robust. Further, the simulations used to evaluate the performance of our statistical test take account for potential sampling bias (see Supplementary Text 5 for details). As we use a probabilistic approach, we expect our model will gain power as sampling of HP/LP viruses improves in the future.

Mutations that increase the fitness of the HP phenotype, for example through permissive/compensatory effects, are expected to be favoured in a HP genetic background, and therefore occur repeatedly in HP lineages. Mutations that occur at nodes preceding the emergence of pCS-defined HP clusters (e.g. mutations that are present in the immediate LP ancestors and within a HP cluster) may represent permissive changes that facilitate the subsequent LP−HP evolutionary transition. Alternatively, mutations that occur immediately after the emergence of an HP cluster (after a pCS has arisen) may be more likely to be compensatory changes. Other statistical methods, such as that described in Kryazhimskiy et al.[57], use a phylogenetically informed approach to detect positive epistasis between pairs of codons in the absence of prior biological information[57]. However, it will not inform on the timing of occurrence for sequential mutations, and thus will not provide information on whether changes are permissive or compensatory. A valid approach for estimating the timing of occurrence for sequential mutations is the reconstruction of ancestral states under a time-scaled analysis. However, we were not able to infer with sufficient resolution the relative timing of occurrence of most HAPMs under our time-scaled phylogenies, in part due to the longer branches leading to some of the HP clusters.

Some of the HAPMs identified here were independently identified in other studies, either in vitro or in silico, as being functionally associated with the HP phenotype (Table 1)[42,45,46,48–50]. Nonetheless, even after our study, which was designed to be statistically conservative, it is likely that the number, biological roles, and genomic distribution of mutations involved in HP adaptation is underestimated. Notably, the HAPMs described here are not shared among the H7NX and H5NX virus subtypes, yet they commonly occur within the same genome segments (Table 1), suggesting that the evolution of the HP phenotype may interact with other well-studied biological processes, such as antigenic escape (determined by the HA protein) and the replication efficiency of viral genomes,

e.g. at lower temperatures (regulated by PB2). Our findings are also in agreement with previous works proposing that the acquisition of a pCS may require coadaptation of the HA and other viral proteins[58].

Some of the mutations occurring close to antigenic sites may be indirectly associated with differential patterns of immunity in varied host populations, defined by complex factors such as animal age and sex distributions, previous exposure to infections/vaccines, or differences in host genetic diversity[59]. Interestingly, mathematical modelling has suggested that the sporadic population dynamics of HP strains may be determined, at least in part, by their linkage to one or more specific antigenic variants, whose frequencies are driven by the dynamics of host immunity, rather than by mutability[24,60]. Furthermore, the emergence of the HP phenotype has been often associated with virus adaptation from wild to domestic bird host populations[51]. The strength of this association is difficult to assess in nature given the substantial bias (in both the literature and sequence databases) towards AIV infections and HP outbreaks in poultry versus those in wild birds. However, using our model of mutation−trait association (see 'Methods', subsection 'Testing for genotype-to-phenotype associations' and Supplementary Text 5), we found no association between any of the detected HAMPs (or any other mutation) and the host species (domestic versus wild bird). This suggests that the parallel mutations identified here are not directly linked to transmission among domestic bird populations, although high population densities together with a limited genetic variability within the host (as occurs in intense farming) may facilitate virus transmission. If acquiring a pCS does indeed reduce AIV transmission fitness in natural systems[34], then it is interesting to speculate that both intensive farming and the occurrence of secondary mutations (HAPMs) may independently act to facilitate the spread of an HP strain; the latter by potentially increasing the transmission fitness of strains carrying a pCS, and the former by lowering the threshold host density required to sustain transmission in a population.

The results we present are derived from computational analysis and are thus strictly predictive. Thus, further experimental validations are needed to understand how the HAPMs actually contribute to the evolution of the HP phenotype, for example, by stabilising protein structure, altering host−virus or virus−virus inter-molecular interactions, or by combining additively to increase virulence. However, such experimental validation may take several years to complete especially if HP AIV gain-of-function experiments will be required[61]. In the meantime, we should continue to exploit fully the information revealed by parallel molecular evolution, which represents a powerful 'natural' replicated experiment. This situation further highlights the importance of systematic sampling and whole genome sequencing of both HP viruses and their immediate LP ancestors, both prior and during AIV outbreaks.

Lastly, we propose that the set of HAPMs we predict to be associated with HP clusters selected here (or any improvement to this set obtained in the future), may, with sufficient validation and testing, function as an early detection system capable of highlighting lineages that pose a greater risk of evolving the HP phenotype in the future, and can function as a platform for future experimental validation. Importantly, the mutational panel identified here could benefit AIV control even before the mechanistic and functional actions of the mutations described are elucidated. We suggest that circulating AIV strains in domestic and wild bird populations that do not contain a pCS site, but which exhibit a combination of the HAPMs identified here, are potential targets for more focussed and targeted genomic and epidemiological surveillance.

## Methods

We developed an analysis approach to detect parallel genetic changes associated with the LP-to-HP evolution of AIV (summarised in Fig. 1). For convenience, we refer to these parallel genetic changes as HAPMs (HP-cluster-associated parallel mutations). We begin by identifying HP clusters from large-scale AIV phylogenies. Next, we use phylogenetically informed subsampling to create smaller datasets suitable for detailed analysis. The subsampled alignments are then analysed using phylogenetic ancestral state reconstruction to identify convergent mutations that might be associated with HP clusters. Associations between the mutations identified and the HP phenotype (i.e. HAPMs) are formally tested under our model. The reduced list of HAPMs is further analysed to assess their potential functional and selective relevance using the following criteria: (i) evidence of evolving under positive selection, (ii) predicted to be stabilising to protein structure, and (iii) known or predicted functional relevance derived from structural analysis and literature survey. Finally, to explore whether HAPMs might be permissive or compensatory in nature, we assessed the temporal occurrence of mutation using a molecular clock analysis. Our analysis is explained below.

**Large-scale phylogenetic analysis**. We downloaded from the Influenza Virus Database all available nucleotide sequences (>80,000 sequences) corresponding to complete AIV genome sets for the H7NX ($n \approx 3000$) and H5NX ($n \approx 10,000$) subtypes, from all hosts (excluding humans) and geographical regions. Genomes from non-avian hosts included mammals (mostly equine and swine influenza A viruses), environmentally derived sequences (usually from avian faecal sampling) and ND/NA (host species Not Determined/Not Available). Non-avian isolates comprise <15% of the large-scale datasets; most of them representing sporadic cross-species transmissions, with exception of the established equine H7NX lineage[62]. Sequences were excluded if (i) they were >200nt shorter than the full coding sequence length for each genome segment, (ii) they were identical, or (iii) they contained >10% ambiguities. The main eight ORFs (PB2, PB1, PA, HA, NP, NA, MS and NS) were identified and extracted. The number of H7NX ORFs were PB2 = 1856, PB1 = 1787, HA = 2217, PA = 1760, NP = 1513, M1 = 1682 and NS1 = 1231, and for the H5NX viruses were PB2 = 3554, PB1 = 3817, PA = 3421, HA = 5650, NP = 3230, M1 = 2824 and NS1 = 3690. Short ORFs for PB1-F2, PA-X, M2 and NS2 were excluded from analysis, as their short-length precluded the molecular evolutionary analyses undertaken here. For the NA segment, all available NA sequences were downloaded for both the H7NX and H5NX virus subtypes (1739 sequences for H7NX and 4202 for H5NX).

Sequence datasets were aligned using MAFFT[63] and preliminary large-scale phylogenies for each genome segment were estimated using RAxML under a general time reversible nucleotide substitution model with gamma-distributed among-site rate variation (GTR + G). Branch support was assessed using a bootstrap approach with 100 replicates[64]. We used the large-scale phylogenies to identify all HP lineages. HP strains were identified by (i) detecting >1 basic residue insertions within the consensus cleavage site in the HA protein alignments[11] and (ii) by confirming HP status with case reports in the literature[5,11]. Sequences in other alignments were labelled HP/LP according to their status in the HA tree.

**Phylogenetically informed subsampling**. To render analyses computationally feasible, we subsampled the large-scale datasets in a phylogenetically informed manner. Firstly, clusters of HP sequences were further sub-selected under the following criteria: (i) they are monophyletic and well supported (bootstrap value > 80) in the large-scale phylogenies of HA and other genome segments, (ii) they contain >2 HP sequences for all genome segments, (iii) they have identifiable immediate LP ancestors, and (iii) they emerged independently from other HP clusters, and are thus distinct in time and space. Eight HP clusters were selected from 10 H5NX and 15 H5N1 previously identified HP lineages[11], and nine HP clusters were selected from 24 H7NX HP lineages[11]. HP sequences were excluded from our analyses if they did not form well-supported clusters, had no identifiable immediate LP ancestors, or if sequences for all genome segments were not available. For LP sequences, subsampling was undertaken so as to retain the mutational pathways along internal tree branches that lead from LP ancestors to HP clusters, whilst reducing overall alignment sizes. All non-avian viruses were excluded from all subsampled alignments, as viruses the rate and nature of molecular evolution can vary in different hosts[62].

For the H7 and H5 HA datasets we retained (i) the selected HP clusters, (ii) their immediate LP ancestors in the tree, (iii) up to 250 randomly selected LP reference sequences to maintain overall tree topology, and (iv) a group of ancestral sequences for rooting purposes. In total, 313 sequences were sampled for H7 and H5 (Supplementary Data 1). For the NA datasets, we extracted all sequences corresponding to the NA types that are associated with the HP phenotype (for H7NX: N1 = 136 sequences, N3 = 367, N7 = 196 and N9 = 18, and for H5NX: N1 = 2053 subsampled to 200, N2 = 367, N7 = 757, N3 = 78, N8 = 93 and N9 = 223). We then merged the individual NA subtype alignments to create two global NA alignments (for H7NX: NX = 919 and for H5NX: NX = 1146 subsampled to 600). Finally, to illustrate the genetic diversity of NA, we extracted from the large-scale datasets all NA sequences of the isolates present in the HA subsampled alignments (211 for H7NX and 150 for H5NX) (Supplementary Fig. 4).

AIVs of different subtypes undergo genome reassortment, especially within the internal segments[23]. In order to account for this, when inferring the evolutionary history of internal genes, we used a single phylogeny/alignment that includes all AIV subtypes (not just viruses belonging to the H5 and H7 subtypes). In this way, we maximise our chances of including those phylogenetic nodes that are closely related LP ancestors to HP clusters, irrespective of AIV subtype. This procedure aims to (i) reduce estimation error, (ii) increase accuracy of phylogeny reconstruction, and (iii) ensure the most closely related strains to each HP lineage are included. Consequently, for each internal segment (PB2, PB1, PA, NP M1, NS1), we constructed a consensus sequence for each HP cluster and used blastn to identify closely related sequences belonging all AIV subtypes other than H5NX and H7NX, and retained the top ten hits for each search. Sequences from these hits were added to the merged internal segment datasets (both the H5NX and H7NX viruses, including the selected HP strains). The resulting final number of sequences in each internal segment dataset was: PB2 = 514, PB1 = 541, PA = 485, NP = 503, M1 = 443 and NS1 = 471.

Each dataset was re-aligned using MAFFT[63] and alignments were screened for recombination using the RDP, GENECOV and BOOTSCAN components of the RDP3 package[65]. No evidence for intra-segment recombination was detected by three or more methods under a Bonferroni-corrected $p$ value < 0.001. The subsampled alignments were analysed using jModelTest2 to identify well-fitting substitution models[66]. For each alignment, trees were estimated using RAxML under the GTR + G model and bootstrapped with 100 replicates[64]. Prior to tree estimation, HA alignments were edited to remove the cleavage site, whilst regions with poor alignment scores were removed from the NA alignments. Alignment sizes for the subsampled datasets are provided in Supplementary Data 1. The varying sizes and/or missing HP clusters in some alignments are due to the lack of complete genomes for all strains in the public database.

**Detecting candidate HAPMs co-occurring across HP clusters**. To identify candidate HAPMs within AIV genomes, all mutations (deletions, insertions and non-synonymous variable sites) present in ≥1% of the sequences in each subsampled alignment were coded as a discrete taxon trait and subjected to ancestral state reconstruction on the subsampled RAXML trees, using the maximum likelihood framework implemented in the 'ape' R package[67]. The resulting reconstructions were inspected visually to identify candidate HAPMs shared by multiple HP clusters, indicating convergent evolution within the HP phenotype. We identified sites as candidate HAPMs if: (i) they occurred within two or more HP clusters, (ii) they were present in >60% of the sequences within the HP clusters (i.e. the mutation is dominant within a given HP population), and (iii) they were absent from internal nodes that do not lead to HP lineages. Thus, candidate HAPMS may occur on LP terminal branches across the phylogenies, but are not generally fixed within LP lineages (Supplementary Data 2).

**Testing for genotype-to-phenotype associations**. We developed an evolutionary model to statistically test for associations between binary phenotypic traits and amino acid states in a phylogenetically informed manner. Our method serves the same purpose as a $\chi^2$ test, but correctly incorporates the phylogenetic history of the trait and phenotype of interest; failing to account for these phylogenetic correlations can lead to erroneous false positives[20,68]. The model assumes that phenotypes and amino acid states are discrete traits that change along the phylogeny. Given an alignment, its phylogeny, and a set of binary values for each sequence, the method tests if a specific amino acid state (candidate HAPM) is associated with the HP phenotype. This is repeated for all amino acid states and all codons. The coupling parameter $\lambda$ is used to measure the strength of association between an amino acid state and the phenotype. The null hypothesis of zero association is represented by $\lambda = 1$, and can be rejected in favour of a positive association ($\lambda > 1$) using a likelihood ratio test (LRT). The type I and II error rates of this test were evaluated using simulations. We applied this test to our subsampled alignments and used it to select those mutations that are significantly associated with the HP phenotype, thus classifying them as HAPMs. To avoid overrepresentation of the long-lived H5N1 HP viruses[22], the lineage was further reduced in size by collapsing it to its most basal sequences. A full mathematical description of the model and simulation details are provided in Supplementary Text 5.

To explore if the detected HAPMs might represent permissive or compensatory change, we performed a time-scaled analysis coupled with reconstruction of ancestral states using BEAST v1.8.4 [69]. The temporal signal in all RAxML trees was assessed using TempEst v.1.5, by plotting root-to-tip genetic distance against sampling time[70]. As expected, this showed strong temporal signal in all trees, suggesting a strict molecular clock is adequate for these data. From the subsampled alignments, we estimated the posterior probability of amino acid states occurring at tree nodes that are ancestral and immediate to HP clusters, using a discrete trait evolution model. For this, time-scaled phylogenies were estimated using a SRD06 nucleotide substitution model, a strict molecular clock model and a Bayesian skyline coalescent tree prior[69]. Partitions were generated for each site, and ancestral states/transition rates for each mutation of interest were estimated using an asymmetric continuous time Markov chain model[71]. Two MCMC runs were computed for $100 \times 10^6$ states until convergence was reached. Maximum clade

credibility (MCC) trees were summarised using TreeAnnotator after removing 10% of the runs as burn-in[69]. Finally, we note that the presence of HAPMs in HP viruses that were not included in the subsampled alignments could be used to test our analysis pipeline. Therefore, to determine if selected HAPMs were present in these HP sequences, we re-estimated the full mutational history of HAPMs by mapping amino acid changes onto the tips of the large-scale ML phylogenies generated in step 1 (Supplementary Data 3).

**Testing for diversifying positive selection**. The dN/dS ratio (also denoted $\omega$) is the ratio of non-synonymous substitution rate per non-synonymous site to the synonymous substitution rate per synonymous site. It measures the selective pressures acting on a protein coding sequence[72]. Site-specific models (M1a/M2a in CODEML[73] and SLAC, FEL and MEME in Datamonkey[74]) estimate the proportion of codons that are evolving under purifying selection ($\omega < 1$), nearly neutral/neutral evolution ($\omega = 1$), and diversifying positive selection ($\omega > 1$). Branch-site models (BSa/nBSA in CODEML and bsREL in Datamonkey) allow $\omega$ to vary along both branches and among sites simultaneously, making it possible to identify lineages and specific codons that have evolved under positive selection through time[73,74]. In all cases, nested models are tested using the $\chi^2$ approximation to the LRT statistic, and are further subjected to Bonferroni multiple correction for multiple testing.

Using the site-specific models, we calculated the proportion of sites evolving in each $\omega$ category and noted if candidate HAPMs were identified as PSS. We used the branch-site models to detect positively selected HP clusters and associated mutations. The subsampled alignments for each genome segment were used in the analyses, except for the mixed NA subtype alignments (Supplementary Table 1). Since different dN/dS-based methods differ in their statistical performance and on the framework used for parameter estimation[72,75], we undertook a conservative approach and only report sites or HP cluster as being positively selected if they were identified as such by at least two different methods (see Supplementary Table 1). Finally, the output of the DEPS algorithm[38] was compared with our results, and the proportions of codons in each evolutionary class were confirmed using the FUBAR method[31]. The LRT results and lists of all PSS identified using CODEML are provided in Supplementary Data 4.

**Inferring mutation effects on protein stability and function**. Frequently, mutations that may render beneficial phenotypes may also be deleterious at a molecular level, by altering protein structural or folding stability[35]. We therefore undertook structural analyses to predict, based on existing structural data and the literature, if the HAPMs identified in previous steps could have possible implications for protein structure and stability, and/or have any potential functional impacts (see references in Table 1). We modelled protein 3D structures from existing structural data in the PDB database (https://pdb101.rcsb.org/), using representative HP sequences from our datasets as queries. For H7, we used the crystal structure of isolate A/H7N2/New York/107/2003 AIV (PDB ID 3M5G), whilst for H5 we used the crystal structure of isolate A/Vietnam/30262III/04 AIV (PDB ID For the PB2/PB1/PA complex), the high-resolution structure of the A/H5N1/duck/Fujian/01/2002 virus (PDB 6QPF) was used as a template[76]. For all structures, we used the I-TASSER server[77] to generate models with the mutations of interest, which were visually compared to PDB files for validation. Finally, to predict which HAPMs might possible affect the stability of protein structures, we used the PIPS (Phylogenetic Inference of Protein Stability) program[35]. This program implements an informed Bayesian approach and estimates, for each amino acid state, a value of free energy change (ddG) in terms of the thermodynamic stability of an unfolding/refolding protein. A statistical model is used to calculate the probability of free energy change along phylogeny branches in a manner analogous to the computation of nucleotide substitution models, and thereby computationally predict the occurrence of mutations altering protein stability[35]. Negative ddG values are predicted to be stabilising for protein structure. Values close to zero are predicted to have little effect or to be only weakly stabilising, whilst positive values are predicted to destabilise protein structure[35].

**Reporting summary**. Further information on research design is available in the Nature Research Reporting Summary linked to this article.

## Data availability
All sequence data supporting the findings of this study are publicly available from GenBank with accession numbers provided in Supplementary Data 1.

## Code availability
The Julia source code for our model (compatible with Windows and Linux) is available at: https://github.com/michaelgoldendev/trait-evolution.

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

## Acknowledgements

M.E.-Z was supported by an EMBO Long Term Fellowship (ALTF376-2017) during this work, and is currently supported by Leverhulme Trust ECR Fellowship (ECF-2019-542). M.G., J.T. and O.G.P. are supported by the EU's Seventh Framework Programme (FP7/2007-2013)/European Research Council (614725-PATHPHYLODYN). B.G. is supported by the 2017 Universities of Academic Excellence Scholarship Program of the Secretariat for Higher Education, Science, Technology and Innovation of the Republic of Ecuador (ARSEQ-BEC-003163-2017) and by Universidad San Francisco de Quito. J.R.K. and L.C. are supported by the Wellcome Investigator award (200835/Z/16/Z). The Wellcome Centre for Human Genetics is supported by grant 203141/Z/16/Z. T.A.B. is supported by the Medical Research Council (MR/L009528/1). We thank Dr. Sergey Kryazhimskiy for his constructive comments on our model, and Dr. Sergei Kosakovsky Pond for his valuable advice on the methodology for detection of positive selection.

## Author contributions

M.E.-Z. and O.G.P. designed the research. M.E.-Z. performed the research. M.E.-Z. and O.G.P. analysed data. M.G. developed the statistical model and analysed data. B.G. and J.T. analysed genomic data. J.R.K. and L.C. analysed structural data. T.A.B. supervised structural data analysis. M.E.Z. wrote the manuscript, with comments from all authors.

## Competing interests

The authors declare no competing interests.
