## [Peer Review File · Nature Communications]

Reviewer #1:

Remarks to the Author:

Your paper about which amino acid mutations in avian influenza genes are correlated with the acquisition of the multibasic cleavage site in the HA protein (that gives rise to the Highly Pathogenic phenotype) is interesting and contains potentially important results which could be validated experimentally.

I think the good points of the paper are (i) use of a statistical method on public H5 and H7 data (now spanning many years) which accounts for the phylogeny, and thus avoids the 'founder effect' fallacy which plagues many recent machine-learning or association test type papers, and (ii) attempting to show computationally that the identified sites have some phenotypic effect, by considering positive selection measures and protein structure stability measures.

The manuscript generally reads well but I worry that there are some underlying issues with the execution and analysis, which the you should at least discuss or attempt to fix.

Regarding the methods - you claim your statistical phylogenetic association method is new - which I agree it is, but in the details not the overall concept. For example earlier work and software (by others) such as BayesTraits does allow for correlated trait association on phylogenies, its just that the earlier work will only do two binary traits, whereas this manuscript describes a binary trait + 20 state amino acid model.

You also don't say very much about whether the identified mutations are permissive or compensatory - however I believe that there could be some analysis which you could do - you could generate time-scaled phylogenetic trees, and then using the ancestral reconstructions (which will be a by-product of the trait modelling on the trees) comment upon whether the mutation looked like it occurred pre or post the low path to high path transition (with probabilities etc). Or alternatively explain why this is not suitable.

Furthermore earlier work by Kryazhimskiy et al (2011) on epistatic interactions in HA of human influenza could be considered in this paper. Kryazhimskiy et al have a model which accounts for correlations of mutations in time - it would be interesting to include this in the discussion (or even run their model) as a Kryazhimskiy-like model might help with evaluating whether a mutation is permissive or compensatory.

Regarding the data - sequences with H5 and H7 subtypes only are used, and the low or high path character is modelled onto the corresponding trees of the HA segment and internal segments. It is not clear what you are doing with the NA segment - you seem to be making separate giant cross NA-subtype trees (one for the NAs corresponding to the H5s and one for the those corresponding to the H7s), but report results from just the N2s or N3s in the supplementary. Apart from overall illustration, I think it best to analyse the NA subtypes individually (including initial alignment) because they are diverse - you might already be doing this but it is not clear in the manuscript.

For the internal segments, although you state that within each high path cluster of HA there is no further reassortment (which is not true at all for clade 2.3.4.4 within the named C8 cluster anyway), and you include LP ancestors for the individual clades, I understand these LP ancestors to be only from H5 or H7 subtypes (line 104-106). But what if the ancestral sequences to the clades of internal segments dont have a H5 or H7 subtype; which means that closely related sequences to the ancestors to the clades of interest in the internal segments might not be in the phylogeny, but they could still be within Genbank - and this means that detecting selection and timing when a mutation appeared will be more problematic than necessary (due to long branches). To illustrate this I attach a single simple

NJ tree of all the PB2 sequences mentioned in the H5 and H7 files of the supplementary (save about 3 which were only partial sequences); and the tree is roughly coloured by subtype. This roughly indicates that in PB2, clades are not really split into H5 and H7 subtypes, and it would therefore make sense to do the analysis the whole of the PB2 alignment and not split it to generate separate PB2 trees of H5 and H7 subtypes. (And I'm fairly sure that the same effect will be seen in the other internal segments also). You'd probably have to have more than a binary trait for LP/HP though - e.g. H5-LP, H5-HP, H7-LP, H7-HP etc ?

Besides the general points above, please see the below for various detailed points:

INTRODUCTION

2nd paragraph - line 39 or so - perhaps mention where in the HA0 protein the cleavage site occurs ? because it is in the equivalent place in H5 and H7 (between HA1 and HA2) (e.g. typically around position 340 if measuring from the first M, or whatever coordinate system seems appropriate)

The point is that although the individual insertion events are distinguishable from each other, they are all quite similar

(couple of tiny pedantic points which might not even be correct)

line 44 - Do you mean just 'Parallel evolution' rather than 'Parallel molecular evolution' because further on in the sentence you have genotype and phenotype ?

line 47 - mutation rate vs substitution rate ?

line 56 - put the sentence the other way around ? as in Except for RNA structural studies.. (ref 18), little is known..

METHODS

Figure 1 - although Figure 1 does show the overall strategy of the method, it would perhaps be useful to put another tree with two correlated traits marked on it and maybe an equation or matrix etc instead of the dots for stage 4 and possibly 5 ? and where are the clusters on this figure ? (line 76)

(Large scale phylo analysis)

line 86 - how many (or approximately how many) were downloaded, and what is their temporal range ?

(e.g. over 10,000 sequences were available spanning 1950-2018 or whatever the values are)

- small points (possibly important later):

'all hosts' - you mean you have human, other mammals and avian in the data set ?

'all coding' - so you are relying on the correct annotations for e.g. PB1-F2 and PA-X ? rather than just downloading complete genome sets containing full length segments and re-extracting the regions yourself ? or are you just using the largest ORF in each segment ? so this would mean that your segment 7 and 8 alignments for tree building are the smaller M1 and NS1 rather than the full length segment ?

(line 94 - general point for future consideration, not for this work - would PhyCLIP help with the cluster identification ?)

* Subsampling

The subsampling procedure seems very sensible, and allows the retention of informative sequences. However, you are treating the internal segments with H5 partners and H7 partners separately when

I'm not sure this is the best thing to do (see above, and/or explain why you think your approach is OK, because it is not obvious from the trees).

line 111 - you are not aligning different NA subtypes separately ? hence why you perceive 'hypervariable regions' - if you do them separately, you will find a variable stalk region ? and I think this might be important (or a thing to test) because short-stalk NA is associated with adaptation to chickens, and their appearance may well also be associated with appearance of the HP insertion, so you probably do want to have that region included in the individual subtype alignments and not delete it.

line 114 - good that you are providing details in the supplementary, but can you please just indicate in the text what the sizes approximately were maybe something like - 'alignment sizes ranged from X(HA)-Y(PB1)'

Also, about the accession numbers - in one file these correspond to the protein coding regions, and in the other file these correspond to the nucleotide sequences. Please can you harmonise.

* Detecting HAPMs

The selection of important sites sounds fine, and the initial ancestral state reconstruction on the trees with R package ape (on RAxML trees ?) and BEAST (discrete trait) is also good.

Suggest that you add in the text some words to the effect that the number of different amino acids per site (and therefore the size of the discrete rate matrix) was 2-4 (or whatever) - the point being that you are not choosing sites which are so variable that you would need the full 20x20 amino acid matrix ?

* Testing associations

This evolutionary model method sounds OK, but I think it worth emphasising how it is different from programs like BayesTraits which already have a Pagel model for correlated binary characters and takes as input a set of phylogenetic trees.

You are using a binary trait only for the LP/HP phenotype and a larger matrix (20x20) for the amino acid sites, which makes your model different.

I think some depiction of the Q matrix in figure 1 might be helpful as well.

Do you have the option of using a standard amino acid substitution model e.g. JTT, BLOSUM, WAG, FLU etc as the submatrix rather than LG2008 ?

As your implementation is new you could perhaps include a link to the code here as well as the supplementary ?

* Testing for diversifying selection

These set of selection tests are good, was there much difference between the BSA in CodeML and BSREL in HyPhy ? And approximately how many PSS are identified ?

(a range will do but the point is that if it is 100s and the alignment is only 100 then this is the too many variable sites vs sequences problem).

* Protein structures

The protein structure modelling methods seem generally good.

line 189 - Although the bat polymerase structure is the only one with all 3 polymerases, do you think the results would be very different if you used just 2 polymerases, or the full RNP ?

RESULTS

line 214-224 - your depiction & numbering of the different clusters is OK, but can you please correlate

some of them to the H5 naming scheme ?

as in presumably C8(H5N1) is Gs/Gd, and C8(H5NX) is 2.3.4.4 ? There seems some discrepancy between what you have written in the main text, figure legends and supplementary. In the main text you refer to clade 2.2 whereas I think you mean 2.3.4.4 (e.g. see <https://www.ncbi.nlm.nih.gov/pmc/articles/PMC4548997/>).

I'm not sure there is an official H7 scheme though.

Figure 3 - Presumably you have not performed the NA selection analysis on the giant cross subtype trees of NA ? Please be clearer on what alignments / trees are used for what (in the text).

line 271 - 277: (permissive vs compensatory) I think you could (i) use alignments of internal segments with more than only H5 or H7 subtypes (or at minimum just combine the H5 and H7 datasets), and (ii) use time scaled trees and trait mapping to help answer this (see general points above). However, also bear in mind for non HA segments, that you might not be calculating LP/HP evolution coupled with other amino acid evolution, but rather some combination of LP/HP evolution + reassortment propensity coupled with the other amino acid evolution.

line 292-294: I think it would be helpful to briefly explain what the different selection tests, or types of selection test are trying to measure.

line 316: probably just expand on 'antigenic pocket' a little ? - something like - X antigenic regions have been identified in H7 HAs, and site 143 is within antigenic pocket A...

DISCUSSION

No other specific comments, but please see the general comments particularly regarding LP ancestors (with non H5, H7 subtypes) and detecting permissive vs compensatory mutations.

Reference mentioned:

Kryazhimskiy, S., Dushoff, J., Bazykin, G. A., & Plotkin, J. B. (2011).

Prevalence of epistasis in the evolution of influenza A surface proteins. *PLoS Genetics*, 7(2).

<https://doi.org/10.1371/journal.pgen.1001301>

Reviewer #2:

Remarks to the Author:

This study provides the evolutionary analysis of sequences of avian influenza viruses (AIV) to investigate parallel molecular evolution and its relationship with virulence evolution. Coding data was downloaded from a database and authors applied the traditional phylogenetics pipeline (phylogenetic tree reconstruction, ancestral sequence reconstruction and estimation of selection through dN/dS) for its analysis. The study includes an interesting statistical test to evaluate association between mutation and phenotype. Indeed, it also includes a basic analysis of the influence of some amino acid mutations on the protein stability. I found the manuscript well written. My comments are mainly focused on methods and significance of the study, which I think could be improved. Overall, in my opinion, the study is interesting but it does not provide new data, some of the methods applied can be improved and results are highly surprising. Please find below my specific comments.

Major comments

Some information about the analyzed data is not clearly described. Which genomic regions are analyzed? How many sequences (and from which countries) are analyzed? Which data is included in every subsample? This information is crucial and should be clearly presented in the main document.

The "large-scale phylogenetic analysis" includes phylogenetic analysis of all the data, but this data was not tested for the presence of recombination. Only the subsamples were tested for recombination. Next, the subsamples were selected according to the large-scale phylogenetic tree, but this tree could be incorrect if at the global level recombination is present and ignored, see for example, Schierup MH, Hein J. Consequences of recombination on traditional phylogenetic analysis. *Genetics*. 2000, 156: 879-91.

If the large-scale phylogenetic tree is incorrect the selection of subsamples could be biased.

What about if the large-scale data (without the subsampling) is analyzed (I know that it may not be computationally feasible)? Could the results be different to those based on subsamples? What is the effect of subsampling on the results?

Since the data is coding, why not perform phylogenetic inferences using codon substitution models with tools like codonphym1? According to several studies (i.e., by Anisimova et al), codon models are much more informative (and can lead to more accurate inferences) than non-codon DNA substitution models.

Why a strict molecular clock was assumed?, it is common to find evolutionary accelerations/decelerations in the evolution of viruses.

It is well-known that all ancestral sequence reconstruction (ASR) methods present considerable error in the inferences. This error increases with simplistic substitution models (i.e., as indicated above, applying a non-codon DNA substitution model when the data could better fit with a codon substitution model) and with other aspects such as the amount of genetic diversity present in the data. Should ASR be repeated using coding substitution models? How the ASR error could affect the results of this study? Note that errors in the ancestral reconstruction of states from many sites could dramatically affect the predicted protein stability.

Several tools (i.e., codeml or methods implemented in datamonkey) have been applied to estimate dN/dS. Have they provided similar estimates?. If not, which method could be considered as more trustable?

Section "6. Inferring mutation effects on protein stability and function" (and also Fig1, point 6) is confusing, there is not any analysis in the present study to estimate/infer protein function. In this section the text describes analyses to explore the effect of mutations on protein stability, but not analyses to explore the effect of mutations on protein function. Protein stability and protein function are different issues (large or small stability does not imply large or small activity) and are analyzed with different tools or experimentally. For example, the effect of mutations on protein function is often explored using docking analyses (calculating and comparing the binding free energy between "substrate - wild type protein" and between "substrate - mutated protein"). So I recommend rewrite the text and focus only on what was really done.

Indeed, in section 6 basically the study seems to apply only a kind of homology modeling approach to obtain structures from sequences, where mutations were just incorporated by replacing and using PyMOL (line 193). This is a procedure that can provide large error in the resulting structure because ignores hydrophobic/electrostatic/steric/solving/etc interactions among protein sites. After performing homology modeling (i.e., with Modeller; Sali lab) an energy minimization and/or a molecular dynamics of the protein structure is important to accommodate the mutations in the protein structure environment and in all to obtain a more realistic protein structure.

Authors indicate that dN/dS estimation methods are conservative in detecting sites under positive selection in some circumstances (lines 298-299), an argument convenient to support the results of the present study. However, an opposite effect can occur where nonsynonymous substitutions between amino acids with similar physicochemical properties (i.e., Gly <-> Ala or Ser <-> Thr) can be considered as neutral substitutions (they may not alter the protein function, hence wild type and mutated states present a same fitness) when they are traditionally considered as a signature of positive selection.

In general, interpreting dN/dS in terms of selection can be problematic when confounding mutations with substitutions and when comparing data from a single or from different populations. Caution must be taken when using estimated dN/dS to evaluate selection.

Apart of those points, I did not clearly find in the manuscript what is the global (all sites) dN/dS and how dN/dS varies along sequences and over time.

In Results, section "Structural and protein stability analyses". After the first paragraph, the text basically does not present results from the present study (as mentioned before, this study does not present any analysis of protein function), instead it uses information from the bibliography to associate mutations with protein function. First, this is confusing because bibliography is not a result from this study. Second, bibliography is based on proteins with sequences probably different to those from the present study, and processes such as allostery can occur affecting protein function. The influence of a mutation on protein function is not as simple as evaluating the function of other proteins presenting that particular mutation because usually the protein function is a consequence of many sites (different sequences with a same mutation can present different function). As indicated in a previous comment, protein function should be explored in the wet lab or with experiments like docking considering natural substrates.

The results presented in this study (i.e., parallel mutations affecting virulence) have been not tested with experiments performed in the wet lab, so this study only provides predictions. This is mentioned at any point in the manuscript but I think it should be further highlighted.

Minor comments

The GTR+G was selected to perform phylogenetic inferences. Why this substitution model? (i.e., why

not GTR+G+I?)

Suggestion about "RDP3 package". Perhaps authors used the last version (RDP4) and may want to update the text and reference.

"Two MCMC runs were computed for 100x10⁶ states or until convergence was reached". Does it mean that there were cases where convergence was not reached (those sampling 100x10⁶ states)? If yes, can one believe in those estimates?

Could be detected signatures of coevolution among sites (i.e., affecting allostery)?

Reviewer #3:

Remarks to the Author:

Summary:

The authors present a phylogenetic analysis and develop a novel statistical test to interrogate mutations associated with evolution of an HA polybasic cleavage site during avian influenza virus evolution. The evolution of highly pathogenic avian influenza viruses is of significant concern to human and animal health, and the authors present an interesting and novel analysis that I believe will be of interest to readers of Nature Communications. Of particular interest is the development of a novel, phylogenetically-informed statistical test for associations between discrete traits, with possible applications to other virus and trait evolution systems, and the possibility of using inferred associated mutations for improved surveillance.

The manuscript is generally well-written, and I also very much appreciate the extensive supplemental material and data, and the public availability and inclusion of source code with the submission. Before publication, I have some concerns regarding the impacts of sampling, population structure, and host status on the results. I also think that the paper would benefit from careful attention to reformulating the goals, expectations, and conclusions of the study. Specific comments are below.

Major revisions:

Uneven sampling and population structure

1. Population structure: How do the authors expect their statistical test to perform in the presence of population structure and uneven sampling across clusters/clades or geographic regions? Most avian influenza clades are not evenly sampled, and many clades likely have mutations fixed simply from population structure. Furthermore, sampling is not even across disparate geographic locations. Does their method account for this? If not, how will this impact their findings? It may be helpful to include an additional sensitivity analysis that assess the robustness of their method to skewed geographic and clade sampling.

2. Uneven sampling of high and low path viruses: The authors rightly note that high-path viruses are over-sampled compared to low-path viruses. Although this is a bias that is inherent to avian influenza sampling, the authors should devote more care to how this may impact their method and findings. In particular, DTA is used to infer the mutation profile and the pathogenicity profile at internal nodes. DTA is well-known to be biased when the sampling intensity of a deme is not equivalent to the population size of that deme. Because high-path viruses are much more likely to be sampled than low-path viruses, it raises the concern that the reconstruction of the high-path phenotype on internal nodes is biased, such that high-path viruses will be more likely to be inferred ancestors. Therefore,

inferences of linkage between a mutation and the high-path phenotype could be erroneously inferred, and the concern is that if low-path viruses were sampled more frequently, you would see that those inferred co-occurring mutations occur on high-path ancestors as frequently as they on low-path ancestors. While obtaining more low-path sequences isn't reasonable or possible, performing 2 additional analyses would help.

a. Subsample the tree to contain a 50:50 mix of low-path and high-path sequences, and repeat the analyses. Ideally, the authors would perform ~10 or so different subsamplings in which the tree has an equitable ratio of high-path to low-path sequences, and report whether their findings hold. I expect that some of the results will change, but it would be good to report on the robustness to these sampling differences.

b. For simulations related to the new binary trait test, it would be helpful to have sensitivity/recall values when the simulated data have similar sampling biases to the natural sequences (e.g., more HP sequences sampled than LP and then test when there is or isn't an association). This would provide a good estimate of how robust the method is to sampling differences on data with known association values.

3. Sampling and host species: The authors find a number of mutations that are associated with pCS evolution that also elicit putative mammalian-adapting phenotypes. For example, PB2 627K, a well-known marker of mammalian adaptation is detected and associated with pCS evolution, as are multiple mutations in HA located in the receptor binding domain. As currently written, this raises the concerning possibility that mutations that promote pCS evolution might also enhance the likelihood of cross-species transmission. However, the authors do not discuss what fraction of sequences are from human cases, or how the inclusion of human sequences may impact their results. Human cases are likely vastly oversampled compared to poultry cases, and are also likely skewed towards high-path infections. This raises the concern that skewed sampling has led to identifying human-adapting mutations rather than pCS compensatory/permmissive mutations. For these mutations that have putative mammalian-adapting phenotypes, are these identified in clades with high counts of sequences from humans? How do the authors findings change if performed on avian-only sequences? Given that pCS evolution is generally thought to emerge during transmission in poultry, the rationale for including all hosts in the analysis is not clear. A few additional analyses would improve this aspect of the paper:

a. An additional, supplemental figure and corresponding discussion in the results of an analysis when only avian sequences are used. Especially because the authors are interested in using this method for surveillance, this would be useful for knowing how their method would work on avian surveillance samples, while also controlling for identifying host-adapting mutations.

b. Additional information in the methods to explain the inclusion of all host species in the analysis. For example, were ferret samples, which likely derive from laboratory transmission studies, included?

4. Hitchhiking: Given that there is no within-segment recombination in influenza (as shown by the authors in their recombination scans), how are the authors differentiating between HA HAPMs that are co-occurring and compensatory/permmissive, vs. those that have hitchhiked with the pCS?

Rationale for stabilizing mutations

5. The rationale for looking for stabilizing mutations is not entirely clear. The requirement for an acid-stable HA for human transmission and H1N1pdm virus evolution is well-documented, but specifically refers to the requirement that HA remains intact during endosomal acidification in different host species. Do pCSs render the HA less stable in the endosome/in the presence of low pH? Because pCS acquisition is generally thought to evolve in birds, rather than in humans, why might enhanced HA stability be favored? Perhaps this paper might be helpful

<https://journals.plos.org/plospathogens/article?id=10.1371/journal.ppat.1002398>

6. It would be helpful if the authors dedicated more discussion to how acquisition of a pCS would be expected to impact evolution of other genes. Although having compensatory in HA and NA makes intuitive sense, it is not immediately clear which phenotypes in other proteins would be selected and why. Do the authors expect pCS acquisition to impact tissue tropism and interaction with different host machinery? Temperature differences? Why might stability be a selected trait in other, non-HA proteins? Addition of these expectations in the introduction would greatly increase clarity of the authors' hypotheses and expectations. The authors should also include their expectations for how such mutations would appear in their phylogenetic reconstruction.

Reformulating goals and conclusions

7. The study is inherently cross-sectional, making it difficult to infer causality. I think that this is fine, and the study is still useful and interesting, and that this work goes a long way to producing findings that could be causally tested in the lab. However, the paper would be better served by making it clear that they have identified mutations associated with pCSs, which could be associated for any number of reasons (population structure, host-specific sampling, compensatory/permmissive phenotype functions, hitchhiking), rather than attempting to assign causality. The paper might be stronger if the message was reframed to describe host-specific, antigenic, and other correlated mutations without expecting these mutations to be related to the pCSs.

8. The section of positive selection is a bit confusing. The authors lay out rationale for expecting to find compensatory mutations under positive selection, and then find positive selection at only a subset of sites. However, the authors then make the argument that this is expected, given that these mutations might occur before or after pCS evolution. The authors should spend a bit of time clarifying what their expectations are in both the introduction and the results. Additionally, in the Methods, section 5, "Testing for positive selection", it would be nice to include a bit more information about what each of these methods does, why they were specifically chosen, and what the expected results would be. For example, how do you expect the branch and branch-site models to perform differently for this data, and which is more relevant or a more realistic model for your data? Were any sites identified in some methods and not others, and why would that be expected? Which types of sites might be best detected by any of these particular methods?

9. The authors report p-values of 0.0, which does not make sense statistically. These p-values should be reported as less than some value rather than 0.

Minor revisions

1. Line 93, the authors write that they confirm high-path virus status by case reports. How is this done for wild bird outbreaks? Additionally, how were sequences classified into domestic and wild bird, especially for duck sequences?

2. Lines 125-126, point iii, the wording here confused me a bit. Does this mean that the HAPMs were not observed on a single other internal branch in the whole phylogeny, or that they were not observed on any other internal branch preceding a different high-path cluster? Additionally, this means that convergence could not be detected?

3. The bat polymerase is highly divergent from other influenza A virus polymerases. A discussion of the caveats of modeling amino acid mutations onto this polymerase should be included.

4. Figure 2 could be simplified greatly by replacing the maps in panels A and C with smaller tables positioned in the whitespace of the corresponding phylogenies where each table would contain the information shown in tooltips and the pCS sequences could be shown aligned to each other along with

the consensus pCS reported for each region in the text. It was difficult to connect the consensus pCS in the text with those reported in the figure, in the current design.

5. HAPMs shared by high-path clusters were identified visually, but as a sanity check and for reproducibility of the analysis it would be nice if these associations could be algorithmically identified.

6. On line 258, a potential control for the method is described based on high-path clusters from the full phylogeny that did not make it into the subsampled phylogeny. This was surprising as the methods for the subsampling suggest that all high-path clusters are included.

7. Statements in the results section could be strengthened by replacing unspecific wording like "several", "most", "strong", or "a variety" with specific counts and/or percentages. Similarly on lines 39 and 40, are the "over ten" and "over fifteen" pCSs equal to 11 and 16, respectively?

8. Lines 140-142 read "Additionally, we re-estimated the full evolutionary history of these HAPMs by mapping the amino acid changes onto the large-scale ML phylogenies generated in step 1." The meaning of this isn't clear. How were these amino acid changes mapped back onto the ML phylogenies, which in theory may have different topologies, tips, and internal nodes, and why was this performed?

Grammatical and typographical changes:

1. In supplemental methods, a typo reads, "nT and nN represents the frequencies of the non-target (T) and target traits (N) states, respectively." I believe that it should read, "nT and nN represents the frequencies of the target (T) and non-target traits (N) states, respectively."

2. Lines 259 and 281, the "if" should be removed, so they read, "we investigated whether these lineages..." (259) and "Consequently, we investigated whether some HAPMs..." (281).

Reviewer #1 (Remarks to the Author):

(1) Regarding the methods - you claim your statistical phylogenetic association method is new - which I agree it is, but in the details not the overall concept. For example, earlier work and software (by others) such as BayesTraits does allow for correlated trait association on phylogenies, it's just that the earlier work will only do two binary traits, whereas this manuscript describes a binary trait + 20 state amino acid model.

We fully acknowledge that our model builds on previous methods, and citations were included in the Methods Section "Testing for genotype-to-phenotype associations". In response, to make this clearer, we have now added a paragraph to the Discussion section, in which we compare our model to previously published methods, and discuss the advantages and disadvantages of our approach. We also include two new references.

The following text has been added to the Discussion Section (L477-L487):

"Our new statistical model for mutation-phenotype association reformulates the test of Bhattacharya et al²⁰ as an evolutionary-informed phylogenetic model that measures the strength of association between traits. Our test uses a probabilistic approach to account for the potential founder effects (e.g. hitchhiking events) that arise from shared common ancestry, whilst maximising the statistical power to detect associations. Our new model shares similarities with some discrete trait evolution models^{69,70}, however these only consider two binary traits, whereas our test associates a binary trait (HP/LP) with multiple states (20 amino acids). Thus, our model captures the potential dependence between a trait and a target amino acid using a single parameter, rather than requiring a large number of parameters to represent each possible transition separately. This simplicity results in easily-interpretable hypothesis testing, straightforward inference using maximum likelihood, and lower estimation variance."

(2) You also don't say very much about whether the identified mutations are permissive or compensatory - however I believe that there could be some analysis which you could do - you could generate time-scaled phylogenetic trees, and then using the ancestral reconstructions (which will be a by-product of the trait modelling on the trees) comment upon whether the mutation looked like it occurred pre or post the low path to high path transition (with probabilities etc). Or alternatively explain why this is not suitable. Furthermore, earlier work by Kryazhimskiy et al (2011) on epistatic interactions in HA of human influenza could be considered in this paper. Kryazhimskiy et al have a model which accounts for correlations of mutations in time - it would be interesting to include this in the discussion (or even run their model) as a Kryazhimskiy-like model might help with evaluating whether a mutation is permissive or compensatory.

In our original manuscript, we did indeed assess the temporal occurrence of HAPMs (to determine their permissive or of compensatory nature), by undertaking a time-scaled reconstruction of ancestral states analysis using BEAST, as described in Methods section 4 "Testing for genotype-to-phenotype associations" (L200-218). Specifically, we estimated time-scaled trees and further reconstructed ancestral states for the sites of interest (HAPMs) to assess statistical support. In Figure 3 we show (for a representative number of HAPMs) the probability of a given mutation occurring on the branch or branches immediately ancestral to the defined HP clusters. Given the phylogenetic resolution permitted by our current data, it was not possible to determine, for most of the HAPMs, whether these mutations are permissive or compensatory (Results Section, L324-334). "The likely cause of this is a sampling bias towards HP isolates collected during outbreaks, resulting in an under-representation of the immediate LP ancestors of HP clusters". Thus, we cannot determine the exact relative timing of appearance of most HAPMs to the appearance of the pCS (which confers virulence). This limits our discussion on whether if the HAPMs are permissive or compensatory.

Regarding the methodology proposed by Kryazhimskiy et al (2011), it is not clear if the reviewer is expecting to see an analysis comparing HAPMs with HAPMs, or one that uses the pCS as a 'leading trait' to detect associated HAPMs (trailing sites). The latter is already explicitly tested within our model framework, as our model statistically tests the association between the HP phenotype (as defined by a pCS) and other mutations occurring along different nodes of a tree (Supplementary Information, Supplementary Text 5). We engaged in personal correspondence with Prof Kryazhimskiy to explore this matter. We concluded that (i) performing epistasis analyses using the Kryazhimskiy model is redundant, as it uses a similar statistical approach as our own, (ii) the method is not suitable for detecting permissive or compensatory changes, as it is designed to detect pairs of related sites (e.g. mutations) without *a priori* knowledge, and (iii) the Kryazhimskiy method was not optimized for AIVs and thus there might be important differences in sensitivity and specificity when applying it to AIV datasets.

We have clarified further in the main text how we initially selected candidate HAPMs, and the time-scaled approach we used to estimate their timing of occurrence. We have also extended our discussion and modified Figure 3 to explicitly represent this, leading to changes in the following sections:

Methods (L78-92)

Methods Section 3 (L167-178) and 4 (L200-218)

Results Section (L324-359)

Discussion Section (L470-484)

(3) Regarding the data - sequences with H5 and H7 subtypes only are used, and the low or high path character is modelled onto the corresponding trees of the HA segment and internal segments. It is not clear what you are doing with the NA segment - you seem to be making separate giant cross NA-subtype trees (one for the NAs corresponding to the H5s and one for the those corresponding to the H7s), but report results from just the N2s or N3s in the supplementary. Apart from overall illustration, I think it best to analyse the NA subtypes individually (including initial alignment) because they are diverse - you might already be doing this but it is not clear in the manuscript.

As the reviewer guesses, we did indeed analyse the NA subtypes individually, but our methods description was too concise and thus didn't explain our approach clearly enough. To address this, we provide a step-by-step breakdown of our methodology:

- 1) For both the H7NX and H5NX viruses, all available NA sequences were downloaded (1739 sequences for H7, and 4202 for H5 in total). HP sequences were flagged according to the annotate large-scale HA trees.
- 2) For each virus subtype, individual NA types associated to the HP phenotype were extracted from the large-scale datasets and were further aligned using MUSCLE. Individual trees were estimated using RAXML and analysed to detect mutation patterns within single NAs (Table 1, Individual NA alignments).
- 3) In order to compare mutation patterns among different NAs (as HP isolates from both virus subtypes differ in the NA segment), global alignments were built by concatenating the previously described individual alignments. Datasets were further aligned using MUSCLE, deleting regions with poor alignments scores and trees were estimated using RAXML (Table 1, global alignments).
- 4) Finally, from the original large-scale datasets, sequences corresponding to the HA subsampled alignments were obtained using an *in-house* script (a total of 211 for H7NX and of 150 for H5NX we retrieved) (Table 1, subsampled global alignments matching HA). Datasets were re-aligned and trees were estimated using RAXML. These were used to generate Supplementary Figure 3.
- 5) General mutation patterns associated to HP clusters were visually identified in all datasets, and were further analysed under our pipeline.
- 6) As a note, Supplementary File1 includes only accession numbers for the subsampled datasets matching HA.

Table 1.

Subtype H7NX	Subsampled global alignments matching HA	Individual NA alignments	HP cluster	Subtype H5NX	Subsampled global alignments matching HA	Individual NA alignments	HP cluster
N1	45	136	C2	N1	34	2053 (subsampled to 200)	C8
N2	10		LP	N2	74	757	C1, C2 C3, C7
N3	84	367	C6, C7, C8	N3	9	78	C4
N4	4		LP	N4	3		
N5	0		LP	N5	2		
N6	2		LP	N6	2		
N7	34	196	C5	N7	1		
N8	4		LP	N8	4	93	C5
N9	28	223	C1	N9	5	18	C6
All Ns associated to HP (global alignment)	211	919		All Ns associated to HP (global alignment)	134	1146 (subsampled to 600)	

We have added a more detailed explanation in the Methods section (L139-146):

"For the NA datasets, we extracted all sequences corresponding to the NA types that are associated with the HP phenotype (for H7NX: N1=136 sequences, N3=367, N7=196 and N9=18, and for H5NX: N1=2053 subsampled to 200, N2=367, N7=757, N3=78, N8=93 and N9=223). We then merged the individual NA subtype alignments to create two global NA alignments (for H7NX: NX=919 and for H5NX: NX=1146 subsampled to 600). Finally, to illustrate the genetic diversity of NA, we extracted from the large-scale datasets all NA sequences of the isolates present in the HA subsampled alignments (211 for H7NX and 150 for H5NX) (Supplementary Figure 4).

(4) For the internal segments, although you state that within each high path cluster of HA there is no further reassortment (which is not true at all for clade 2.3.4.4 within the named C8 cluster anyway), and you include LP ancestors for the individual clades, I understand these LP ancestors to be only from H5 or H7 subtypes (line 104-106). But what if the ancestral sequences to the clades of internal segments don't have a H5 or H7 subtype; which means that closely related sequences to the ancestors to the clades of interest in the internal segments might not be in the phylogeny, but they could still be within Genbank - and this means that detecting selection and timing when a mutation appeared will be more problematic than necessary (due to long branches). To illustrate this, I attach a single simple NJ tree of all the PB2 sequences mentioned in the H5 and H7 files of the supplementary (save about 3 which were only partial sequences); and the tree is roughly coloured by subtype. This roughly indicates that in PB2, clades are not really split into H5 and H7 subtypes, and it would therefore make sense to do the analysis the whole of the PB2 alignment and not split it to generate separate PB2 trees of H5 and H7 subtypes. (And I'm fairly sure that the same effect will be seen in the other internal segments also). You'd probably have to have more than a binary trait for LP/HP though - e.g. H5-LP, H5-HP, H7-LP, H7-HP etc?

This is an astute and important observation (reiterated in comment 25 from this author) It is indeed better to use a single phylogeny/dataset that includes all AIV subtypes for each internal gene. By doing so we can (i) reduce estimation error, (ii) increase accuracy of phylogeny reconstruction, and (iii) ensure the most closely related strains to each HP lineage are included, thereby increasing power. In response, we devised a modified and improved methodology that addresses the issue of reassortment within the internal segments of the H5 or H7 HP clusters. We then repeated our analyses, as follows (L147-154):

- 1) We constructed a consensus sequence for each HP cluster of both virus subtypes, and used blastn to identify the most closely related sequences belonging to *all other* AIV subtypes, other than H5NX and H7NX.
- 2) We included the top ten hits (i.e. the most closely related sequences to each HP cluster that were non-H5 or non-H7), in new datasets.
- 3) We merged the alignments of both virus subtypes (e.g. PB2 H5NX+H7NX) for all internal segments, and included the new sequences from other subtypes identified in step 2. The merged datasets were realigned and used to estimate RAXML trees that were further used for reconstruction of ancestral states under ML to evaluate the mutational patterns mutations previously identified.

As expected, internal segment trees were not structured by subtype. Despite this, all HP clusters are monophyletic and well supported with high bootstrap values in all trees (i.e. the data shows no reassortment *within* HP clusters).

There are two exceptions: C1 H7N9 is split into two clusters (C1.1. an C1.2) in all internal segment trees, and the C8 H5N1 cluster is split into two clusters (C8.1. an C8.2) in the PB2, PA and NP trees (Supplementary Figures 2 and 3). Many HP clusters have ancestral sequences from different virus subtypes other than H5NX or H5NX, though the phylogenetic placement of these are not well supported. Reconstruction of ancestral states using ML revealed that mutations in candidate HAPMs are consistent with our previous results (Supplementary File 2) (**Results section L287-300**).

Our results are consistent with no evidence for reassortment among HP clusters. As correctly pointed out by the reviewer, an exception for this is the H5NX C8 (Cluster 2.3.4.4), that is long established in the domestic bird population. This mode of evolution differs significantly from the one observed for other HP clusters. In order to acknowledge the differences between C8 and other HP clusters, we collapsed C8 to its most basal sequences for all further analyses, as stated in Methods Section 4 (**L195-196**).

Finally, we re-ran our model using these new datasets. In general, we observed an improvement in the robustness of our model, were significant HAPMs previously scored under our model where retained. Three HAPMs that were previously scored as the least significant now were ranked as not significant and thus were removed from the revised results:

HAPM	Segment	Subtype	old p-value	new p-value
D209N	NS1	H7NX	0.03	0.08
G154D	PB1	H7NX	0.03	0.14
I451T	PB2	H5NX	0.03	0.14

Notably, although HAPMs are not observed in non-H5/H7 sequences, the inclusion of non-H5/H7 sequences significantly improved the association test statistics for many sites listed in Table 1. This is likely due to a lack of substitutions towards the target amino acids in non-H5/H7 sequences. This lack of substitution, somewhat counterintuitively, provides a greater evidentiary weight to the occurrence of substitutions towards the target amino acid in H5 and H7 sequences with HAPMs. Table 1 in the main text was modified according to our new results. Figure 4, Supplementary Figures 2 and 3, and Supplementary File 2 were also updated according to our new results.

(5) 2nd paragraph - line 39 or so - perhaps mention where in the HA0 protein the cleavage site occurs? because it is in the equivalent place in H5 and H7 (between HA1 and HA2) (e.g. typically around position 340 if measuring from the first M, or whatever coordinate system seems appropriate). The point is that although the individual insertion events are distinguishable from each other, they are all quite similar (couple of tiny pedantic points which might not even be correct)

This has been clarified as follows (**L33-35**):

"The HA cleavage site occurs at the C-terminal domain, within a single arginine residue (Arg-343 in H1 numbering), and is highly conserved among all influenza A viruses, including AIVs (Ref 11)."

(6) line 44 - Do you mean just 'Parallel evolution' rather than 'Parallel molecular evolution' because further on in the sentence you have genotype and phenotype?

"Parallel evolution" describes the independent emergence of similar traits from the same ancestral state, or from different ancestral states. We use the term "Parallel molecular evolution" to define this process at a molecular level, specifically, the genetic pathways that lead to the emergence of a same phenotype in different viral subtypes (e.g. defined in Ref 14 and Zakon. 2002. Brain Behav Evol. 59:250-61).

We have now clarified the definition in the main text as follows (**L45-47**):

"We use the term 'parallel molecular evolution' to define this process at a molecular level, specifically referring to the convergent or parallel genetic pathways that lead to the emergence of same phenotype in different viral subtypes (Ref 15)."

(7) line 47 - mutation rate vs substitution rate?

In this sentence, we refer to the mutation rate, which is the "rate at which errors are made during replication of the viral genome" (e.g. Peck et al, 2018). This contrasts with the substitution rate, which is the rate at which mutations become fixed in a population.

We have clarified the main text as follows (**L47-48**):

"RNA viruses exhibit high rates of mutation (i.e. the rate at which errors are made during the virus genome replication)"

(8) line 56 - put the sentence the other way around? as in Except for RNA structural studies. (ref 18), little is known.

The sentence has been modified as suggested (**L59**):

"Little is known about the genetic predisposition of AIV to acquire a pCS (except for the role of RNA structure at the cleavage site region itself¹⁹)"

(9) Figure 1 - although Figure 1 does show the overall strategy of the method, it would perhaps be useful to put another tree with two correlated traits marked on it and maybe an equation or matrix etc instead of the dots for stage 4 and possibly 5? and where are the clusters on this figure? (line 76)

Figure 1 and its corresponding legend have been modified as suggested.

(10) line 86 - how many (or approximately how many) were downloaded, and what is their temporal range? (e.g. over 10,000 sequences were available spanning 1950-2018 or whatever the values are)

We are sorry if this was not made clear in the main text. We have added a sentence in the Methods Section to clarify this (L95-97).

(11) 'all hosts' - you mean you have human, other mammals and avian in the data set?

As the large-scale trees were mainly used to detect global evolutionary patterns among the AIVs of interest (H7NX and H5NX subtypes), sequences from all host, including non-avian, were used in the initial analysis. Once we identified global evolutionary patterns and selected the HP clusters of interest, we excluded all non-avian sequences for the subsampling. It has been shown that viruses evolving within different hosts have independent rates of molecular evolution, and thus this could be a confounding factor for phylogenetic analyses (Worobey, et al. 2014). In any case, from the metadata files, we observe that the contribution of non-avian viruses to the large-scale datasets represent <15% of all sequences, and that most of these are likely non-established lineages arising cross-species transmissions, with exception of the HP equine viruses (Worobey, et al. 2014).

We have clarified this in the main text as follows (L95-102):

"We downloaded from the Influenza Virus Database all available nucleotide sequences (>80,000 sequences) corresponding to complete AIV genome sets for the H7NX (n≈3000) and H5NX (n≈10000) subtypes, from all hosts (excluding humans) and geographical regions. Genomes from non-avian hosts included mammals (mostly equine and swine influenza A viruses), environmentally derived-sequences (usually from avian faecal sampling) and ND/NA (host species Not Determined/Not Available). Non-avian isolates comprise <15% of the large-scale datasets; most of them representing sporadic cross-species transmissions, with exception of the established equine H7NX lineage²¹."

(L134-135):

"All non-avian viruses were excluded from all subsampled alignments, as viruses the rate and nature of molecular evolution can vary in different hosts²¹."

(12) 'all coding' - so you are relying on the correct annotations for e.g. PB1-F2 and PA-X? rather than just downloading complete genome sets containing full length segments and re-extracting the regions yourself? or are you just using the largest ORF in each segment? so this would mean that your segment 7 and 8 alignments for tree building are the smaller M1 and NS1 rather than the full-length segment?

For this analysis, we downloaded complete genome sets and extracted the main ORFs using an in-house script. We used only the larger ORFs, as most of tools and methods used for the analyses (e.g. reconstruction of ancestral states and detecting positive selection) lose power results when used in short alignments with low sequence diversity (as expected from short ORFs, such as PB1-F2 and PA-X). Moreover, despite the increasing popularity of whole genome sequencing, there is still a large bias in the databases towards larger genome segments (specially HA and NA), resulting in accessory ORFs being even less available than main ORFs (as an example, PA has 1555 sequences available vs PA-X, that has 743 sequences). We also decided to focus on non-accessory accessory proteins that are critical for the main biological functions of the virus. For the M and NS segments, we only used M1 and NS1 ORFs to represent the evolution of the single largest gene within a given genome segment.

We have clarified this in the Methods Section 1 as follows (L104-109):

"The main eight ORFs (PB2, PB1, PA, HA, NP, NA, MS and NS) were identified and extracted. The number of H7NX ORFs were PB2=1856, PB1=1787, HA=2217, PA=1760, NP=1513, M1=1682 and NS1=1231, and for the H5NX viruses were PB2=3554, PB1=3817, PA=3421, HA=5650, NP=3230, M1=2824 and NS1=3690. Short ORFs for PB1-F2, PA-X, M2 and NS2 were excluded from analysis, as their short-length precluded the molecular evolutionary analyses undertaken here."

(13) line 94 - general point for future consideration, not for this work - would PhyCLIP help with the cluster identification?

We thank the reviewer for this advice. There are many tools for cluster identification that are mainly based on pairwise genetic distance, tree structure and branch support, among others. However, in this particular case, visual identification was required, as we were not sure if all HP sequences would form defined lineages and how well supported these would be. Also, HP sequences are not coded within the databases and had to be first identified according to the literature, and by further analysing the alignments in order to identify insertions within the CS.

(14) The subsampling procedure seems very sensible, and allows the retention of informative sequences. However, you are treating the internal segments with H5 partners and H7 partners separately when I'm not sure this is the best thing to do (see above, and/or explain why you think your approach is OK, because it is not obvious from the trees).

Please see our reply to Major Comment 4, in which we state how we have undertake a new analysis to correct for this.

(15) line 111 - you are not aligning different NA subtypes separately ? hence why you perceive 'hypervariable regions' - if you do them separately, you will find a variable stalk region ? and I think this might be important (or a thing to test) because short-stalk NA is associated with adaptation to chickens, and their appearance may well also be associated with appearance of the HP insertion, so you probably do want to have that region included in the individual subtype alignments and not delete it.

Please see our reply for Major Comment 3. We have further clarified the main text accordingly.

As pointed out by the reviewer, a 20-residue deletion within the stalk region of the N1, N3 and N9 associated to some H7NX HP clusters was identified. However, this deletion was not further scored under our model, and therefore, it was not described in the results section. It is known that the frequency of stalk-deletion mutants fluctuate both in time and space both within LP and HP virus isolates of the H7NX and H5NX subtypes (Li et al, 2011). Although some

deletion patterns have been associated with increased virulence experimentally, the occurrence of deletion patterns are likely to be correlated with host switching (from wild to gallinaceous hosts), rather than with the emergence of HP (McAuley et al, 2019; Li et al, 2011).

We have also added the following paragraph in the Results Section (**L317-323**):

"For some H7NX HP clusters, we identified a 20-residue deletion within the stalk region of the N1, N3 and N9 strains (Supplementary File 2). However, this deletion was not scored as a HAPM under our model. The frequency of stalk-deletion mutations is known to fluctuate in time and space among both LP and HP viruses⁴². Although some deletions have been experimentally linked to increased virulence, their occurrence is likely associated with host switching (from wild bird to gallinaceous hosts), rather than with the emergence of the HP phenotype^{42,43}."

(16) line 114 - good that you are providing details in the supplementary, but can you please just indicate in the text what the sizes approximately were maybe something like - 'alignment sizes ranged from X(HA)-Y(PB1)'. Also, about the accession numbers - in one file these correspond to the protein coding regions, and in the other file these correspond to the nucleotide sequences. Please can you harmonise.

Sizes for the subsampled Alignment are now indicated in **L139, L141-142, L144 and L153**. As suggested, the accession numbers in Supplementary File 1 have been all harmonised to represent nucleotide sequences only.

(17) The selection of important sites sounds fine, and the initial ancestral state reconstruction on the trees with R package ape (on RAxML trees?) and BEAST (discrete trait) is also good. Suggest that you add in the text some words to the effect that the number of different amino acids per site (and therefore the size of the discrete rate matrix) was 2-4 (or whatever) - the point being that you are not choosing sites which are so variable that you would need the full 20x20 amino acid matrix?

A BEAST DTA has to infer the substitution rates associated with all possible transitions between the amino acids observed at a given alignment site. This introduces a source of uncertainty. Whereas in our test for association the rates are fixed because we use an empirical rate matrix (with the exception of having to infer a scaling parameter). Our approach also introduces a source of uncertainty: the fact that we permit transitions between amino acids not observed at a given site, however, this is likely to be a smaller source of uncertainty than that associated with a BEAST DTA. Both types of uncertainty will lower power to detect signals of association, however, we believe our approach of using an empirical rate matrix will be less likely to do so. Using an empirical rate matrix also simplifies our analysis, and implies we do not have to perform a computationally expensive marginalisation of rates parameters, which would make our test of association prohibitively slow.

(18) This evolutionary model method sounds OK, but I think it worth emphasising how it is different from programs like BayesTraits which already have a Pagel model for correlated binary characters and takes as input a set of phylogenetic trees. You are using a binary trait only for the LP/HP phenotype and a larger matrix (20x20) for the amino acid sites, which makes your model different. I think some depiction of the Q matrix in figure 1 might be helpful as well.

For all analyses, we used the LG2008 rate matrix. In theory, our model can use any time-reversible 20x20 amino acid rate matrix. We specifically did not to use the FLU matrix because it is derived primarily from human cases of Influenza, whereas all our sequences pertain to Avian Influenza Viruses (see again Worobey et al, 2014). We chose a more general matrix to reduce potential bias at the cost of having lower power to detect associations.

(19) Do you have the option of using a standard amino acid substitution model e.g. JTT, BLOSUM, WAG, FLU etc as the submatrix rather than LG2008?

Not at present. We are currently in the process of integrating the WAG and FLU matrixes as options in the software and hope to make this option available in the future.

(20) As your implementation is new you could perhaps include a link to the code here as well as the supplementary?

We have included the GitHub link in the main text (**L199**).

(21) These set of selection tests are good, was there much difference between the BSA in CodeML and BSREL in HyPhy? And approximately how many PSS are identified? (a range will do but the point is that if it is 100s and the alignment is only 100 then this is the too many variable sites vs sequences problem).

The proportion of sites scored as PSS are listed within the LRT files provided in the Supplementary File 4. For the M1a/M2a test, the proportion of sites are indicated with "p" under the different classes (K1: purifying, K2: neutral/nearly neutral, and K3: positive diversifying selection). For the BSA/BSREL test, these are indicated in the column p1^a, representing the proportion of sites scored under class 2a: positive selection in the foreground branches and purifying selection for the background branches. Numbers and PSS are also provided in the Supplementary File 4.

As expected for AIVs and for RNA viruses in general (Ref 16), global dN/dS (ω) estimates for whole genome segments were <1 in all cases and the overall proportion of PSS detected was low (Supplementary Table 1, Supplementary File 4). For all alignments, the highest proportion of sites was assigned to the strong conservation class ($\omega \ll 1$) and to the neutral/ nearly neutral class ($\omega < 1$), as assessed under all methods. Thus, we did not observe a proportion of sites with ω estimates outside of what would be expected for IAVs, and for RNA viruses in general. We have further confirmed these results by using the FUBAR method (**L242**). For H5, analysis revealed 458 sites evolving under episodic negative/purifying selection, compared to an estimate of 5 sites evolving under episodic positive/diversifying selection, assessed under a posterior probability of 0.9. For H7, 417 sites were scored as evolving under episodic negative/purifying selection, compared to an estimate of 0 sites evolving under episodic

positive/diversifying selection, assessed under a posterior probability of 0.9. This has been clarified in the main text (L374-378).

Regarding BSA and aBSREL, the differences observed where in the number of clusters and sites that were significantly scored, but not in the proportions of sites and in different sites or clusters (Supplementary Table 1). We must note that BSA and aBSREL were used to detect positively selected lineages, whilst global ω estimates and PSS were scored under site models. In an attempt to harmonize our findings, the main results reported in Supplementary Table 1 represent a summary of all analyses, in which all lineages and PSS were scored at least under two different methods.

(22) The protein structure modelling methods seem generally good. line 189 - Although the bat polymerase structure is the only one with all 3 polymerases, do you think the results would be very different if you used just 2 polymerases, or the full RNP?

The results wouldn't change if we were only to use 2 polymerases or the full RNP complex for structural modelling, as the HAPMs detected do not involve sites with direct interaction within the polymerase subunits. The only relevant site within the catalytic domain is 627, that has been extensively characterized. Nonetheless, the structures of the H5N1 influenza A viral polymerase have been very recently determined (Fan et al, 2019). Thus, we have updated Figure 4 and 5 to use the H5N1 model (PDB: 6QPF), instead of the bat flu polymerase, in order to show the position of the HAPMs of interest accurately. There are no changes to our conclusions. As the interactions between the polymerase and remainder of the RNP are not known, we are unable to say if there would be interactions with either the NP or RNA.

The text in methods section 6 (L254-255) and corresponding figures has been adjusted to account for this.

(23) line 214-224 - your depiction & numbering of the different clusters is OK, but can you please correlate some of them to the H5 naming scheme? as in presumably C8(H5N1) is Gs/Gd, and C8(H5NX) is 2.3.4.4 ? There seems some discrepancy between what you have written in the main text, figure legends and supplementary. In the main text you refer to clade 2.2 whereas I think you mean 2.3.4.4 (e.g. see <https://www.ncbi.nlm.nih.gov/pmc/articles/PMC4548997/>). I'm not sure there is an official H7 scheme though.

We have corrected this in the Results Section (L276-283) and Supplementary Figure 1.

"The large-scale phylogeny of H5NX viruses shows a basal clade of H5NX viruses circulating in wild birds worldwide from the late 1960s to the early 2000s (Supplementary Figure 1, Figure 2). This basal clade is ancestral to the established HP H5N1 lineage (named C8 here), that diverged from A/goose/Guangdong /1/1996 (including sub-lineages 7.2 to 2.3.2.1a)²⁸. Since its emergence in China, this lineage has subsequently spread to Asia, Africa, Europe and America and persists until today²⁸. Within C8, there is a distinct group of HP H5NX viruses, forming clade 2.3.4 and its descendants, that circulate in both wild and domestic birds²⁸."

There is no official scheme for H7.

(24) Figure 3 - Presumably you have not performed the NA selection analysis on the giant cross subtype trees of NA? Please be clearer on what alignments / trees are used for what (in the text).

As explained in our reply to comment 3, we performed selection analyses separately on individual NA datasets. No selection analyses were performed on the mixed NA alignments. The main text and Supplementary Table 1 have been modified to make this clear (L235-236):

"The subsampled alignments for each genome segment were used in the analyses, except for the mixed NA subtype alignments (Supplementary Table 1)."

(25) line 271 - 277: (permissive vs compensatory) I think you could (i) use alignments of internal segments with more than only H5 or H7 subtypes (or at minimum just combine the H5 and H7 datasets), and (ii) use time scaled trees and trait mapping to help answer this (see general points above). However, also bear in mind for non-HA segments, that you might not be calculating LP/HP evolution coupled with other amino acid evolution, but rather some combination of LP/HP evolution + reassortment propensity coupled with the other amino acid evolution.

Although we have repeated our analyses following the suggestions made by the reviewer (see Major Comment 4), the conclusions that can be drawn on whether if the HAPMs detected are permissive vs compensatory are still limited (see our response to Major Comment 2). From our new analyses using merged H5+H7+other AIVs for the internal segments, we observed that the pattern of occurrence for candidate HAPM between LP ancestors and HP lineages along the tree did not change, as can be observed in the modified Supplementary File 2. All HP clusters are well supported with no evidence for reassortment among these (Supplementary Figures 2 and 3), even when adding other AIVs to the datasets. Thus, it is not necessary to introduce a correction parameter for reassortment propensity. Moreover, integrating this variable within our model would only render our model overparametrized. Thus, we conclude *"due to reassortment, the internal gene segments of HP cluster strains may originate from AIV subtypes other than H7NX or H5NX. Nonetheless, low phylogenetic support in the internal segment trees reflects that the origin of these segments cannot, in many instances, be determined (Supplementary Figure 1 and 2)."* (L293-296).

(26) line 292-294: I think it would be helpful to briefly explain what the different selection tests, or types of selection test are trying to measure.

The following text has been modified in the Methods Section 5 (L221-243).

"The dN/dS ratio (also denoted ω) is the ratio of nonsynonymous substitution rate per non-synonymous site to the synonymous substitution rate per synonymous site. It measures the selective pressures acting on a protein coding

sequence³¹. Site-specific models (M1a/M2a in CODEML³² and SLAC, FEL and MEME in Datamonkey³³) estimate the proportion of codons that are evolving under purifying selection ($\omega < 1$), nearly neutral/neutral evolution ($\omega = 1$), and diversifying positive selection ($\omega > 1$). Branch-site models (BSa/nBSA in CODEML and bsREL in Datamonkey) allow ω to vary along both branches and among sites simultaneously, making it possible to identify lineages and specific codons that have evolved under positive selection through time^{32,33}. In all cases, nested models are tested using the χ^2 approximation to the likelihood ratio test statistic, and are further subjected to Bonferroni multiple correction for multiple testing. Using the site-specific models, we calculated the proportion of sites evolving in each ω category and noted if candidate HAPMs were identified as Positively Selected Sites (PSS). We used the branch-site models to detect positively-selected HP clusters and associated mutations. The subsampled alignments for each genome segment were used in the analyses, except for the mixed NA subtype alignments (Supplementary Table 1). Since different dN/dS-based methods differ in their statistical performance and on the framework used for parameter estimation^{31,34}, we undertook a conservative approach and only report sites or HP cluster as being positively-selected if they were identified as such by at least two different methods (see Supplementary Table 1). Finally, the output of the DEPS algorithm³⁵ was compared with our results, and the proportions of codons in each evolutionary class were confirmed using the FUBAR method³⁶. The LRT results and lists of all PSS identified using CODEML are provided in Supplementary File 4.

(27) line 316: probably just expand on 'antigenic pocket' a little? - something like - X antigenic regions have been identified in H7 HAs, and site 143 is within antigenic pocket A...

The requested details have been added to the sentence in question (L401-404):

"This is within antigenic pocket A, contributing to the electrostatic environment needed for ligand binding. Five antigenic regions (A-E) have been identified within antigenic pocket A for the A(H1N1) pdm09 virus, with site 143 located within five residues N-terminal to antigenic site A⁵¹."

(28) No other specific comments, but please see the general comments particularly regarding LP ancestors (with non H5, H7 subtypes) and detecting permissive vs compensatory mutations.

We have expanded our discussion according to our new results, please see our replies to comments 2 and 4 for details.

Reviewer #2 (Remarks to the Author):

(1) Some information about the analyzed data is not clearly described. Which genomic regions are analyzed? How many sequences (and from which countries) are analyzed? Which data is included in every subsample? This information is crucial and should be clearly presented in the main document.

As requested, we have further clarified the Methods Section to clearly state sample demographics, number of sequences used before and after subsampling, and the genome regions used, in the following sections.

Methods Section 1. Large-scale phylogenetic analysis (L95-119)
Methods section 2. Phylogenetically-informed subsampling (L121-165)

The criteria used for selecting HP clusters from the large-scale trees is stated in **L123-128**, whilst our approach used for subsampling LP sequences is stated in **L132-135**. Sequence data included in all subsampled alignments is available in Supplementary File 1.

(2) The “large-scale phylogenetic analysis” includes phylogenetic analysis of all the data, but this data was not tested for the presence of recombination. Only the subsamples were tested for recombination. Next, the subsamples were selected according to the large-scale phylogenetic tree, but this tree could be incorrect if at the global level recombination is present and ignored. If the large-scale phylogenetic tree is incorrect the selection of subsamples could be biased.

The large-scale trees were mainly used simply to identify HP clusters. This identification exercise will not be affected by the possibility of within-segment recombination (the HP clusters are well known in the literature and our analysis simply recapitulates their characterisation). As the reviewer states, the subsampled datasets upon which our actual results were based were tested for recombination, and our results are robust to the exact choice of reference database strains. So, there is no risk of bias from potential recombination. Crucially, although recombination has been described for some RNA viruses (Han et al, 2011), homologous recombination within segments seems to play little or no role in the evolution of influenza A viruses (Boni et al, 2008; Boni et al 2010). Furthermore, current methods for detecting recombination are limited to alignment size and length (generally, no more than 500 sequences can be tested). It would be unfeasible to run recombination analyses for the large-scale datasets.

References:

- Han GZ, Worobey M. Homologous recombination in negative sense RNA viruses. *Viruses*. 2011;3(8):1358–1373. doi:10.3390/v3081358
- Boni MF, Zhou Y, Taubenberger JK, Holmes EC. Homologous recombination is very rare or absent in human influenza A virus. *J Virol*. 2008;82(10):4807–4811. doi:10.1128/JVI.02683-07
- Boni MF, de Jong MD, van Doorn HR, Holmes EC. Guidelines for identifying homologous recombination events in influenza A virus. *PLoS One*. 2010;5(5):e10434. Published 2010 May 3. doi:10.1371/journal.pone.0010434

(3) What about if the large-scale data (without the subsampling) is analyzed (I know that it may not be computationally feasible)? Could the results be different to those based on subsamples? What is the effect of subsampling on the results?

No, because of the way we subsampled. We deliberately subsampled in a phylogenetically-informed way, such that all information pertaining to the mutational paths leading to the HP clusters was retained after subsampling. The informed subsampling we undertook was difficult, but worthwhile, because it generated data sets that kept all the information for our specific hypothesis, whilst reducing them down to a size that is computationally feasible. The reviewer would have a point if we had subsampled randomly – we didn't.

We have further clarified our rationale and methodology in Methods Section 2 (**L132-139**) and addressed sampling bias issues in the Discussion Section (**L460-469**).

(4) Since the data is coding, why not perform phylogenetic inferences using codon substitution models with tools like codonphymi? According to several studies (i.e., by Anisimova et al), codon models are much more informative (and can lead to more accurate inferences) than non-codon DNA substitution models.

In brief, for such highly phylogenetically informative alignments such as ours, any reasonably realistic substitution model will give the same overall clustering patterns (see Shapiro et al, 2009 and <https://www.biorxiv.org/content/10.1101/228692v4.full> for the particular case of IAVs). The models we use are standard throughout the field of influenza virus phylogenetics and it would be odd to use a different class of models here. The GTR model is powerful enough to capture main evolutionary processes with sufficient accuracy at a sensible computational cost. Prior to estimating the RAXML trees, subsampled alignments were tested under the jModelTest to find the best fit model, but this was not mentioned in the main text, as it is considered standard procedure and our methods description needed to be concise.

Nonetheless, our phylogenetic analyses include both the GTR+G used for the RAXML trees (**L115, 158-160**), and the SRD06 codon substitution used for the time-scaled trees (**L208**). The ML and the time-scaled phylogenies were consistent (**L333**). Choosing codon substitution models over nucleotide substitution models would only render our analysis overparametrized and computation time would become impractical. Overfitted models actually increase the error in parameter estimates, because instead of reflecting a real biological process, they may be fitting the noise (e.g. Lemey & Vandamme, 2009).

We have further clarified the Methods Section as follows:

“The subsampled alignments were analysed using jModelTest2 to identifying well-fitting substitution models (Darriba et al, 2012).” (L158-160)

References:

Shapiro B, Rambaut A, Drummond AJ. Choosing Appropriate Substitution Models for the Phylogenetic Analysis of Protein-Coding Sequences. *Mol Biol Evol* 2009;23(1):7-9. doi:10.1093/molbev/msj021
Lemey P, S M, Vandamme A (2009) *The Phylogenetic Handbook: A Practical Approach to Phylogenetic Analysis and Hypothesis Testing*. Cambridge: Cambridge University Press.

References added:

Darriba D, Taboada GL, Doallo R, Posada D. jModelTest 2: more models, new heuristics and parallel computing. *Nat Methods*. 2012;9(8):772. Published 2012 Jul 30. doi:10.1038/nmeth.2109

(5) Why a strict molecular clock was assumed? it is common to find evolutionary accelerations/decelerations in the evolution of viruses.

It is well known that influenza virus genomes are very clocklike in their evolution – possibly the most clocklike of any virus known. Hence a strict clock is perfectly reasonable for IAV and consequently often used in the literature. Using a relaxed clock would add estimation variance without a beneficial increase realism. Furthermore, prior to the time-scaled analyses, we assessed the temporal signal for each of the subsampled trees using TempEst v.1.5 to obtain a regression of root-to-tip genetic distance against sequence sampling time. Our results show – as expected – a high regression coefficients for all segments and strong temporal signal. As an example, an estimated (R^2) of 0.99 (H7) and of 1.00 (H5) were obtained for the HA segments. Possible differences in the IAV rate of evolution among hosts is irrelevant because we exclude all non-avian viruses from all subsampled alignments (**L134-135**).

The following sentence has been added to the main text to clarify this (**L202-205**):

“The temporal signal in all RAxML trees was assessed using TempEst v.1.5, by plotting root-to-tip genetic distance against sampling time (Rambaut et al, 2016). As expected this showed strong temporal signal in all trees, suggesting a strict molecular clock is adequate for these data.”

References added:

Rambaut A, Lam TT, Max Carvalho L, Pybus OG. Exploring the temporal structure of heterochronous sequences using TempEst (formerly Path-O-Gen). *Virus Evol*. 2016;2(1):vew007. Published 2016 Apr 9. doi:10.1093/ve/vew007

(6) It is well-known that all ancestral sequence reconstruction (ASR) methods present considerable error in the inferences. This error increases with simplistic substitution models (i.e., as indicated above, applying a non-codon DNA substitution model when the data could better fit with a codon substitution model) and with other aspects such as the amount of genetic diversity present in the data. Should ASR be repeated using coding substitution models? How the ASR error could affect the results of this study? Note that errors in the ancestral reconstruction of states from many sites could dramatically affect the predicted protein stability.

We first note that this view of ASR largely arises from its application to very long evolutionary timescales, such as inferring proteins from earlier geological eras. We are looking at much more recent changes, which are more robustly estimated. The fact that avian influenza virus sequences are longitudinally sampled (i.e. heterochronous sequence data) and contain high levels of phylogenetic diversity further reduces the uncertainty in ASR estimates. Second, a codon substitution model is highly parametric and makes additional assumptions to the method we use here, hence estimates using the former approach will be less robust and have higher variance.

This is why we reconstructed ancestral states under two different frameworks, ML and Bayesian Inference using a time-scaled approach (DTA), using trees that have been inferred under different sequence evolution models, including GTR and SRD06 (see our reply to comment 4 of this reviewer), and obtained comparable results (Figure 3, Supplementary File 2). As mentioned, the ML and the time-scaled phylogenies are consistent (**L334**), and in general well-supported (Figure 2c and 2d). Thus, we believe that these trees robustly represent the evolutionary history of AIVs, and in particular the HP lineages of interest, and are therefore reliable for performing ASR.

Regarding the comments that “ASR dramatically affecting predictions on protein stability”, we only used ASR to detect candidate mutations, which were then further scored as HAPMs using a statistical model and a detailed methodological pipeline (**L79-92**). On the other hand, predictions of protein stability are undertaken using the PIPS program that uses “an informed Bayesian approach and estimates for each amino acid state, a value of free energy change ($\Delta\Delta G$) in terms of the thermodynamic stability of an unfolding/refolding protein, weighted under a phylogenetic background (Ref 37)”, independent of the ASR analyses (**L258-264**).

(7) Several tools (i.e., codeml or methods implemented in datamonkey) have been applied to estimate dN/dS. Have they provided similar estimates? If not, which method could be considered as more trustable?

All methods are comparable as they all rely on measuring the dN/dS ratio to detect positive selection. All of them are ‘trustable’, but differ on their statistical performance given alignment size and sequence diversity, and on the framework used for parameter estimation (ML or Bayesian approaches) (Ref 31,34). Thus, to harmonize our findings, the main results reported in Supplementary Table 1 represent a summary of all analyses, in which all lineages and PSS were scored at least under two different methods. The use of combined methods is expected to reduce detecting false positives (Anisimova et al, 2002), being conservative and ensuring robustness of our results

We briefly explain what the different types of tests measure by adding two additional paragraphs to the **Methods Section 5 (L220-243)**.

References:

- Anisimova M, Bielawski JP, Yang Z. Accuracy and Power of Bayes Prediction of Amino Acid Sites Under Positive Selection. *Mol Biol Evol.* 2002;19(6):950-958.

(8) Section “6. Inferring mutation effects on protein stability and function” (and also Fig1, point 6) is confusing, there is not any analysis in the present study to estimate/infer protein function. In this section the text describes analyses to explore the effect of mutations on protein stability, but not analyses to explore the effect of mutations on protein function. Protein stability and protein function are different issues (large or small stability does not imply large or small activity) and are analyzed with different tools or experimentally. For example, the effect of mutations on protein function is often explored using docking analyses (calculating and comparing the binding free energy between “substrate - wild type protein” and between “substrate - mutated protein”). So I recommend rewrite the text and focus only on what was really done.

We are aware that protein stability and protein function are very different things. But because IAVs are so well studied, we can explore the possible functional implications of the identified HAPMs in terms of virus biology and biochemical properties (Results Section: Structural and protein stability analyses). However, we only speculate on the functional properties of HAPMs based on structural biology and on existing evidence from the literature. These predictions require future experimental validation, as we state.

An example for this is HAPM A674T co-occurring with E627K. We speculate that the acquisition A674T together with other HAPMs in the polymerase complex would render a structurally stable polymerase (Ref 63) that can efficiently replicate the viral genome, whilst the acquisition of a pCS occurring with other HAPMs in the HA, would result in a structurally-stable hemagglutinin that would allow for a systemic spread of the virus in the avian host. Enhanced replication combined with a systemic viral spread would then increase the chances of transmission within and across hosts, as E627K is also a pre-requirement for the adaptation to mammals by enabling viral replication at lower temperatures and facilitating airborne transmission (Ref 45).

On the other hand, the rationale for exploring the possible effects of HAPMs on protein structure and stability rely on the hypothesis that a mutation underlying a beneficial phenotypic change may also be deleterious at a molecular level, for example, by affecting protein structure or folding stability (Ref 37). This applies to all viral proteins. As stated in our reply to comment (6) in this review, predictions on protein stability are undertaken using the PIPS program (L258-264).

To avoid confusion regarding our methodological approach, we have modified the following sections:

Methods section 6. Inferring mutation effects on protein stability and function

Results section “Inferring possible effects of HAPMs on protein structure and function”

The following sentence has been added to the main text to clarify this (L247-250):

“We undertook structural analyses to predict, based on existing structural data and the literature, if the HAPMs identified in previous steps could have possible implications for protein structure and stability, and/or have any potential functional impacts (see References in Table 1).”

(9) Indeed, in section 6 basically the study seems to apply only a kind of homology modeling approach to obtain structures from sequences, where mutations were just incorporated by replacing and using PyMOL (line 193). This is a procedure that can provide large error in the resulting structure because ignores hydrophobic/electrostatic/steric/solvation/etc interactions among protein sites. After performing homology modeling (i.e., with Modeller; Sali lab) an energy minimization and/or a molecular dynamic of the protein structure is important to accommodate the mutations in the protein structure environment and in all to obtain a more realistic protein structure.

We agree with the reviewer that a final step of energy minimization is important for modeling accuracy. However, the models generated for all proteins were generated using the I-Tasser pipeline, which includes an energy minimization step. With the release of the H3N2 and H5N1 viral influenza polymerase structures (Ref 38), we are now able to confidently model directly the position of the mutations identified for the H5N1 viruses. There is >97% identity between the structures of the H7NX polymerase consensus and the H5N1 Fujian polymerase, which also allows us to model these residues with high confidence. To further address the issue raised by the reviewer, we have performed an additional step of energy minimization using the Phenix geometry minimization program for all models. As many of the residues were located in solvent exposed regions, there is no change to our interpretations.

(10) Authors indicate that dN/dS estimation methods are conservative in detecting sites under positive selection in some circumstances (lines 298-299), an argument convenient to support the results of the present study. However, an opposite effect can occur where nonsynonymous substitutions between amino acids with similar physicochemical properties (i.e., Gly <-> Ala or Ser <-> Thr) can be considered as neutral substitutions (they may not alter the protein function, hence wild type and mutated states present a same fitness) when they are traditionally considered as a signature of positive selection. In general, interpreting dN/dS in terms of selection can be problematic when confounding mutations with substitutions and when comparing data from a single or from different populations. Caution must be taken when using estimated dN/dS to evaluate selection. Apart of those points, I did not clearly find in the manuscript what is the global (all sites) dN/dS and how dN/dS varies along sequences and over time.

The conservative nature of dN/dS methods can be demonstrated theoretically and is not just a “convenient” argument. Further, the reviewer appears to erroneously conflate dN/dS with rates of non-synonymous change. If change between two amino acids (e.g. Gly<->Ala) is neutral, then those changes will by definition occur at the same rate as synonymous change, hence the dN/dS ratio = 1 and positive selection is NOT detected. This is exactly the point of dN/dS methods.

We do acknowledge differences between conservative and non-conservative amino acid changes in the discussion section, in which the biochemical properties of the replacing amino acids were used to inform on any steric interferences altering protein structure, stability and folding properties, as well as modifying interactions with other

molecules and/or altering electrostatic environments, based on existing published structural and experimental data. This is discussed in the results section “Inferring possible effects of HAPMs on protein structure and function” (L395-L435).

We agree, however, that “caution must be taken when using estimated dN/dS to evaluate selection”, and we already state this in our manuscript, by saying that dN/dS methods “differ on their statistical performance given alignment size and sequence diversity, and on the framework used for parameter estimation (ML or Bayesian approaches) (Ref 31, 34), thus “we undertook a conservative approach and only report sites or HP cluster as being positively-selected if they were identified as such by at least two different methods (see Supplementary Table 1)” (L238-240). This latter condition renders our methodological approach even more conservative.

In response to the point “I did not clearly find in the manuscript what is the global (all sites) dN/dS”: for all alignments the highest proportion of sites was assigned to the strong conservation class ($\omega \ll 1$) and to the neutral/ nearly neutral class ($\omega < 1$), assessed under all methods. The overall proportion of PSS detected was low, as expected for IAVs and RNA viruses in general (Ref 16). These are reported in Supplementary Table 1, Supplementary File 4. We have further confirmed these results by using the FUBAR method and have clarified this in the main text (L394-403).

In response to the question “how dN/dS varies along sequences and over time”, this variation is exactly what is explored by the branch-site models (BSa/nBSA and bsREL), that we employ here. (Among branch is equivalent to among time.) As we state, these methods “Branch-site models (BSa/nBSA in CODEML and bsREL in Datamonkey) allow ω to vary along both branches and among sites simultaneously, making it possible to identify lineages and specific codons that have evolved under positive selection through time” (L226-229).

To avoid confusion in our methodological approach, we have modified the following sections:

Methods section 5. Testing for diversifying positive selection
Results section “HAPMs associated with diversifying positive selection”

(10) In Results, section “Structural and protein stability analyses”. After the first paragraph, the text basically does not present results from the present study (as mentioned before, this study does not present any analysis of protein function), instead it uses information from the bibliography to associate mutations with protein function. First, this is confusing because bibliography is not a result from this study. Second, bibliography is based on proteins with sequences probably different to those from the present study, and processes such as allostery can occur affecting protein function. The influence of a mutation on protein function is not as simple as evaluating the function of other proteins presenting that particular mutation because usually the protein function is a consequence of many sites (different sequences with a same mutation can present different function). As indicated in a previous comment, protein function should be explored in the wet lab or with experiments like docking considering natural substrates.

Using computational structural modelling to explore the potential effects of mutations is a well-established approach in the literature and is particularly informative for well known viruses such as avian influenza. Thus, as stated in our reply to comment (8) of this reviewer, interpreting our results in the context of published experimental/structural data (which are often experimentally-tested) is a valid approach. In response to the comment that “bibliography is based on proteins with sequences probably different to those from the present study”, we note that AIVs share up to 98% similarity, and some of these experimental studies even use the sequences corresponding to the HP lineages we analyse here, such as 2000 H7N1 HP outbreak in Italy (Ref 66-68), corresponding to HP cluster C2 (for more examples, see References in Table 1). We acknowledge that other processes such as allostery and epistasis can affect protein function, however we do clearly state that experimental validations in future studies are needed to understand how the HAPMs actually contribute to the evolution of the HP phenotype (L548-552). As other reviewers point out, our results are potentially valuable even if the underlying causal mechanisms cannot yet be elucidated. See (L552-560) and other responses elsewhere in this document.

To avoid confusion regarding our methodological approach, we have modified the following sections:

Methods section 6. Inferring mutation effects on protein stability and function
Results section “Inferring possible effects of HAPMs on protein structure and function”

We have also modified the discussion section to emphasize that “the results we present are derived from computational analysis and are thus strictly predictive. our results are predictive” (L517-518), but “with sufficient validation and testing, function as an early detection system capable of highlighting lineages that pose a greater risk of evolving the HP phenotype in the future, and can function as a platform for future experimental validation” (L528-531).

(11) The results presented in this study (i.e., parallel mutations affecting virulence) have been not tested with experiments performed in the wet lab, so this study only provides predictions. This is mentioned at any point in the manuscript but I think it should be further highlighted.

We have made this suggested change. Please see our reply to the previous comment (11) and comment (8) for the changes made.

Minor comments:

(12) The GTR+G was selected to perform phylogenetic inferences. Why this substitution model? (i.e., why not GTR+G+I?)

The GTR substitution model includes the rate heterogeneity parameter modelled as a gamma distribution (G), that accounts for the differences in evolutionary rates between different categories of sites, and would therefore account

for sites evolving at rates equal to zero (Yang et al, 1994). Adding a second model for rate heterogeneity (invariant sites) would be redundant. More importantly, the invariant sites model is logically inconsistent when applied to error-prone RNA viruses, as we can safely assume a priori that the proportion of invariant sites is effectively zero.

References:

- a. Yang Z. Estimating the Pattern of Nucleotide Substitution. J Mol Evol. 1994;39:105-111.

(13) Suggestion about "RDP3 package". Perhaps authors used the last version (RDP4) and may want to update the text and reference.

RDP3 was perfectly sufficient for our requirement of excluding recombinant sequences. RDP4 incorporates additional features not required for our analyses. Please note that one of us (Michael Golden) was a co-author of the RDP4 package (Martin et al, 2015) so we have a good understanding of the features of this software.

(14) "Two MCMC runs were computed for 100x10⁶ states or until convergence was reached". Does it mean that there were cases where convergence was not reached (those sampling 100x10⁶ states)? If yes, can one believe in those estimates?

This means that convergence can be sometimes reached before the 100x10⁶ states. Both MCMC runs reached convergence within this limit, and thus our estimated are robust. We have removed "or" from this sentence for clarity.

(15) Could be detected signatures of coevolution among sites (i.e., affecting allostery)?

Several methods have been developed to infer coevolution between sites (Talavera et al, 2015; Meyer et al, 2019 and Kryazhimskiy et al, 2011). These methods incorporate the phylogeny of sequences to identify pairs of sites evolving under similar evolutionary processes. For example, the Kryazhimskiy et al (2011) method (Ref 71) aims to detect epistasis without *a priori* knowledge. However, as discussed elsewhere in our responses, in the case of HP AIVs, we have prior knowledge that the pCS is considered the 'leading trait', which is then used to detect HAPMs (trailing sites). Thus, detecting signatures of coevolution among sites (PCS and HAPMs) is already explicitly tested within our model framework, as our model statistically tests the association between pCS-defining mutations and other mutations occurring along different nodes of a tree (Supplementary Information, Supplementary Text 5).

Reviewer # 3 (Remarks to the Author):

Uneven sampling and population structure

(1) Population structure: How do the authors expect their statistical test to perform in the presence of population structure and uneven sampling across clusters/clades or geographic regions? Most avian influenza clades are not evenly sampled, and many clades likely have mutations fixed simply from population structure. Furthermore, sampling is not even across disparate geographic locations. Does their method account for this? If not, how will this impact their findings? It may be helpful to include an additional sensitivity analysis that assesses the robustness of their method to skewed geographic and clade sampling. Uneven sampling of high and low path viruses: The authors rightly note that high-path viruses are over-sampled compared to low-path viruses. Although this is a bias that is inherent to avian influenza sampling, the authors should devote more care to how this may impact their method and findings. In particular, DTA is used to infer the mutation profile and the pathogenicity profile at internal nodes. DTA is well-known to be biased when the sampling intensity of a deme is not equivalent to the population size of that deme. Because high-path viruses are much more likely to be sampled than low-path viruses, it raises the concern that the reconstruction of the high-path phenotype on internal nodes is biased, such that high-path viruses will be more likely to be inferred ancestors. Therefore, inferences of linkage between a mutation and the high-path phenotype could be erroneously inferred, and the concern is that if low-path viruses were sampled more frequently, you would see that those inferred co-occurring mutations occur on high-path ancestors as frequently as they on low-path ancestors. While obtaining more low-path sequences isn't reasonable or possible, performing 2 additional analyses would help. a. Subsample the tree to contain a 50:50 mix of low-path and high-path sequences, and repeat the analyses. Ideally, the authors would perform ~10 or so different subsamplings in which the tree has an equitable ratio of high-path to low-path sequences, and report whether their findings hold. I expect that some of the results will change, but it would be good to report on the robustness to these sampling differences. b. For simulations related to the new binary trait test, it would be helpful to have sensitivity/recall values when the simulated data have similar sampling biases to the natural sequences (e.g., more HP sequences sampled than LP and then test when there is or isn't an association). This would provide a good estimate of how robust the method is to sampling differences on data with known association values.

We are acutely aware of the issue raised here by the reviewer, and our analysis pipeline was designed with it in mind. The crucial point is that we are *not* undertaking a typical phylogeography or DTA analysis. In summary, the oversampling of HP strains may lead to a loss of statistical power (as also explained elsewhere in our responses) but not to bias. Thus, our results are *conservative*: there may exist more HAPMS than we can detect using currently available data, but the ones we *do* detect are sound. Improved future sampling of LP viruses would lead to increased power for our method but not change to the mutations we identify here (**L460-469**)

First, note that the HP clusters are robustly defined using a parsimony approach, because the pCS is a powerful synapomorphy. Thus, the lineages we label as HP *are indeed* HP. However, some lineages we label as LP (i.e. those immediately ancestral to a HP cluster) may be wholly or partly HP. That's because of the undersampling of LP lineages. Mutations on these branches will not be associated with the HP phenotype, thereby statistically weakening our test. The net effect of any biased sampling is therefore to make our test *statistically conservative*. This is good - we can be confident in the significant associations we do find.

We do use the DTA approach for reconstructing ancestral amino acid states at nodes of interest (**L200-218**). However, biased sampling does not occur *with respect to* the amino acid states, it occurs with respect to HP/LP status. Therefore, the sampling bias cannot create the appearance of a mutation/HP link that is not there, but it can potentially obfuscate the detection of such links. As before, as results are conservative and robust.

We also note that our study already includes a subsampling analysis of the kind proposed. Specifically, we obtained similar inferences when analysing the subsampled and large-scale datasets (Methods Section **L200-218**), Results Section **L348-359**, Supplementary File 3). Overrepresented groups were significantly down-sampled in the subsampled datasets. For example, in the large-scale H5 alignment (n=5650), 97% of the sequences correspond to HP viruses and only 3% to LP viruses (and H5N1 isolates represent 74% of all sequences). Whereas in the subsampled H5N1 dataset (n=313), whilst 43% are HP and 40% to LP (and H5N1 isolates represent 38% of all sequences). This confirms that our model results are not sensitive to very large fluctuations in HP/LP sampling bias, which is exactly the point the reviewer is concerned about. Moreover, "the presence of HAPMs in HP sequences that were not used in the subsampled alignments constitute an independent test of our statistical approach" (**L358-359**).

Finally, we note that our simulation procedure already accounts for HP sampling bias. The HP and LP traits were fixed to the trait frequencies observed in the real data, and amino acid site patterns were simulated by conditioning on the fixed trait patterns. Our benchmarks on these simulated data demonstrate that HP sampling bias does not lead to false-positive associations (see **Supplementary Text 5**).

To clarify these points, we have added a paragraph in the Discussion Section (**L460-469**) and modified the Supplementary Text 5.

(2) Sampling and host species: The authors find a number of mutations that are associated with pCS evolution that also elicit putative mammalian-adapting phenotypes. For example, PB2 627K, a well-known marker of mammalian adaptation is detected and associated with pCS evolution, as are multiple mutations in HA located in the receptor binding domain. As currently written, this raises the concerning possibility that mutations that promote pCS evolution might also enhance the likelihood of cross-species transmission. However, the authors do not discuss what fraction of sequences are from human cases, or how the inclusion of human sequences may impact their results. Human cases are likely vastly oversampled compared to poultry cases, and are also likely skewed towards high-path infections. This raises the concern that skewed sampling has led to identifying human-adapting mutations rather than pCS compensatory/permissive mutations. For these mutations that have putative mammalian-adapting phenotypes, are these identified in clades with high counts of sequences from humans? How do the authors' findings change if performed on avian-only sequences? Given that pCS evolution is generally thought to emerge during

transmission in poultry, the rationale for including all hosts in the analysis is not clear. A few additional analyses would improve this aspect of the paper:

- a. An additional, supplemental figure and corresponding discussion in the results of an analysis when only avian sequences are used. Especially because the authors are interested in using this method for surveillance, this would be useful for knowing how their method would work on avian surveillance samples, while also controlling for identifying host-adapting mutations.
- b. Additional information in the methods to explain the inclusion of all host species in the analysis. For example, were ferret samples, which likely derive from laboratory transmission studies, included?

The large-scale datasets were used to evaluate the global genetic diversity and lineage structure of AIVs, and to identify HP clusters. Thus, all isolates were contained in those initial datasets, including AIV lineages established in mammalian hosts, such as the equine and swine clades. All human-derived isolates and ferret samples derived from transmission studies were excluded as they do not represent natural infections/and or established lineages. After this initial analysis, only avian AIV sequences were retained in the subsampled alignments, and it is from these alignments that our study results are obtained. This has now been clarified in the Methods Section 1 and 2 (**L78-92, L94-119 and L121-165**). Thus, we believe that our observations regarding the association of E627K with the HP phenotype are sustained, and are further supported by previous studies that noted that E627K is rare among H5NX AIVs, but particularly frequent among viruses that have caused human infections (**L428-429**, Ref 7).

(3) Hitchhiking: Given that there is no within-segment recombination in influenza (as shown by the authors in their recombination scans), how are the authors differentiating between HA HAPMs that are co-occurring and compensatory/permisive, vs. those that have hitchhiked with the pCS?

The key (and novel) aspect of our methodology is its focus on repeated, *parallel* evolution, which is possible because the HP clusters have evolved many times independently. It is possible (though improbable) that the same mutation may occur by chance on multiple HP clusters. The probability of this happening is exactly what our new phylogenetic test is evaluating. If mutation X co-occurs with mutation (Y) that is causally-linked to the HP phenotype/genotype, and the co-occurrence of these is observed more often than by chance alone, then, by definition, both mutations X and Y are HAPMs. To explain further:

- We focus on mutation that are present in *multiple* HP clusters and have a positive association with the HP phenotype: Groups of closely related viruses can share mutations just because they share common descent (e.g. *hitchhiking*) (Ref 20). Our phylogenetic model explicitly tests for this by it incorporating the phylogenetic history of both the trait (mutation) and phenotype of interest (HP), as it is known that failing to account for these phylogenetic correlations can lead to erroneous false positives. From a large number of initial candidate mutations, we only identified HAPMs as those occurring across multiple HP clusters with a positive association with the HP phenotype (Table 1 in main text).
- HAPMs resulting from hitchhiking would be expected to be neutrally selective, as detected under dN/dS methods. Permissive or compensatory mutations are expected to be advantageous, at least within the genetic context of the HP phenotype. As expected, a proportion of HAPMs were found to be evolving under positive selection despite dN/dS methods being conservative.
- Potential functional relevance of HAPMs is supported by our structural biology analysis; we found that most of the HAPMs scored are expected to be functionally relevant, suggesting that these may have a biological impact.

We have further clarified in the main text how we initially selected candidate mutations, and then use our model and extensive methodological pipeline to rule out mutations that may co-occur with a pCS by chance alone. We have also extended our discussion and modified Figure 3 to explicitly represent this (as in Reply to Reviewer 1):

Methods (L78-92)

Methods Section 3 (L167-178) and 4 (L200-218)

Results Section (L324-359)

Discussion Section (L470-484)

(4) Rationale for stabilizing mutations. The rationale for looking for stabilizing mutations is not entirely clear. The requirement for an acid-stable HA for human transmission and H1N1pdm virus evolution is well-documented, but specifically refers to the requirement that HA remains intact during endosomal acidification in different host species. Do pCSs render the HA less stable in the endosome/in the presence of low pH? Because pCS acquisition is generally thought to evolve in birds, rather than in humans, why might enhanced HA stability be favoured? Perhaps this paper might be helpful <https://journals.plos.org/plospathogens/article?id=10.1371/journal.ppat.1002398?>

When we say “stabilizing mutations” we are not specifically referring to mutations conferring an *acid-stable* HA. More generally, at the protein structure level, multiple mutations are often required to confer an advantageous phenotypic change. However, a mutation underlying a beneficial phenotypic change may also be deleterious at a molecular level, for example, by affecting protein structure or folding stability (Ref 37). This applies to all viral proteins. In the particular case of the HA and a pCS, not much is known on how a pCS can impact protein structure stability. At least at the level of RNA structure, it is known that nucleotide insertions may destabilize predicted stem-loop structure dramatically, reducing RNA editing efficiency. H5 viruses have potential stem-loop structures in the RNA mapping to the cleavage region, suggesting that a pCS could indeed hinder RNA structure stability to some extent (Ref 18).

Within an evolutionary context, the acquisition of viral proteins with stable structures might confer short-term advantages to HP viruses by giving rise to virions capable of replicating and transmitting, at least within dense host populations. In this scenario, protein stability enhanced by HAPMs would be favoured. Nonetheless, with exception of the H5N1 C8, most HP clusters are short-lived lineages; i.e. they emerge and die-off within a time-span too short to allow establishment. Therefore, the HP phenotype might be only favoured in the short-term to then be eliminated, which is possibly why it repeatedly emerges though time and space.

We have added the following text to clarify this (**L246-250**):

“Frequently, mutations that may render beneficial phenotypes may also be deleterious at a molecular level, by altering protein structural or folding stability³⁷. We therefore undertook structural analyses to predict, based on existing structural data and the literature, if the HAPMs identified in previous steps could have possible implications for protein structure and stability, and/or have any potential functional impacts (see References in Table 1).”

(5) It would be helpful if the authors dedicated more discussion to how acquisition of a pCS would be expected to impact evolution of other genes. Although having compensatory in HA and NA makes intuitive sense, it is not immediately clear which phenotypes in other proteins would be selected and why. Do the authors expect pCS acquisition to impact tissue tropism and interaction with different host machinery? Temperature differences? Why might stability be a selected trait in other, non-HA proteins? Addition of these expectations in the introduction would greatly increase clarity of the authors' hypotheses and expectations. The authors should also include their expectations for how such mutations would appear in their phylogenetic reconstruction.

It is well known that a pCS itself does affect tissue tropism, as the insertion of multiple basic residues within the cleavage site of the HA is recognized and cleaved by intracellular subtilisin-type proteases ubiquitously expressed in many tissues. Consequently, HP viruses can spread systemically within susceptible avian hosts. On the other hand, the cleavage site in the HA of LP viruses can only be recognized by the trypsin-like proteases expressed only in the respiratory tract. Thus, the systemic replication of LP AIVs is limited (**L35-37**, Ref 12).

Nonetheless, predicting the effects of epistatic interactions across viral genomes at a molecular level using computational tools is a complicated task, and without experimental validation, we have no guaranteed way of determining how the acquisition of a pCS is expected to impact evolution of other viral genes. Thus, we can only speculate on the functional properties of HAPMs based on structural biology analyses and on the existing evidence from the literature. An example for this is HAPM A674T co-occurring with E627K. We speculate that the acquisition A674T in the polymerase complex would render a structurally stable polymerase that can efficiently replicate the viral genome, whilst the acquisition of a pCS occurring with other HAPMs in the HA, would result in a structurally-stable hemagglutinin that would allow for a systemic spread of the virus in the avian host. Enhanced replication combined with a systemic viral spread would then increase the chances of transmission within and across hosts, as E627K is also a pre-requirement for the adaptation to mammals by enabling viral replication at lower temperatures and facilitating airborne transmission (Ref 45).

The flu genome contains so few genes that all must work together in a highly-coordinated fashion and mutations in virus proteins may have pleiotropic effects. Our understanding of these virus protein/protein interactions (direct and indirect) is so incomplete that it's productive to hypothesise scenarios here. We hope our work will stimulate and direct new experimental work on these questions.

For all HAPMs listed in Table 1, we have extended our discussion on the expected functional impact of these both individually and combined (Results Section: Inferring possible effects of HAPMs on protein structure and function), and on how we expect these to contribute to the HP phenotype is stated in the Discussion Section (**L485-516**). The general expectation of HAPMs impacting viral transmission dynamics are stated in the Discussion Section (**L509-516**). In terms of protein stability, please see our reply to Comment 4 for this review. Our expectations for mutation occurrence patterns in the phylogenetic reconstructions are also stated within the main text (**L470-477**).

(6) Reformulating goals and conclusions

The study is inherently cross-sectional, making it difficult to infer causality. I think that this is fine, and the study is still useful and interesting, and that this work goes a long way to producing findings that could be causally tested in the lab. However, the paper would be better served by making it clear that they have identified mutations associated with pCSs, which could be associated for any number of reasons (population structure, host-specific sampling, compensatory/permmissive phenotype functions, hitchhiking), rather than attempting to assign causality. The paper might be stronger if the message was reframed to describe host-specific, antigenic, and other correlated mutations without expecting these mutations to be related to the pCSs.

The rationale for our study exploits a rather unusual situation: the fact that the pCS has evolved independently many different times, on different genomic backgrounds. It is this crucial notion of parallel evolution that provides our study its power and novelty; parallel evolution is a crucial component of our identification of relevant mutations. Parallel evolution allows our methodology to rule out population structure, host-specific sampling, and compensatory phenotype functions as caused as association between mutations and the HP phenotype. Our approach is not strictly cross-sectional, because time-scaled phylogenetic analyses allow us to resolve to order of genetic events through time.

In our reply to Comment 1 for this review, we also discussed how HAPM identification is independent of host demographic biases, such host-specific sampling. We agree that we cannot strongly assign causality; however, because flu is so well studied we can explore the possible functional implications of the identified HAPMs in terms of virus biology and biochemical properties (Results Section: Inferring possible effects of HAPMs on protein structure and function). We also include caveats that this study is only 'predictive' (**L518**). In response to "the paper might be stronger if the message was reframed to describe mutations within a host-specific and antigenic context", we have discussed this in **L496-516**.

We agree with the reviewer that our results are potentially valuable even if their causal mechanisms cannot yet be elucidated. In part, that's because we propose that the set of HAPMs could "with sufficient validation and testing, function as an early detection system capable of highlighting lineages that pose a greater risk of evolving the HP phenotype in the future, and can function as a platform for future experimental validation" (**L528-531**). The associations alone are sufficient to provide this potential predictive benefit, even if the mechanistic actions of the mutations are unknown (this is the same logic that is used in most machine learning applications, although we'd prefer not to use that buzzword here).

(7) The section of positive selection is a bit confusing. The authors lay out rationale for expecting to find compensatory mutations under positive selection, and then find positive selection at only a subset of sites. However, the authors then make the argument that this is expected, given that these mutations might occur before or after pCS evolution. The authors should spend a bit of time clarifying what their expectations are in both the introduction and the results. Additionally, in the Methods, section 5, "Testing for positive selection", it would be nice to include a bit more information about what each of these methods does, why they were specifically chosen, and what the expected results would be. For example, how do you expect the branch and branch-site models to perform differently for this data, and which is more relevant or a more realistic model for your data? Were any sites identified in some methods and not others, and why would that be expected? Which types of sites might be best detected by any of these particular methods?

The global ω estimates we observed across viral genes were within the range of that expected for AIVs, and for RNA viruses in general (Ref 15). We have now further confirmed these results by using the FUBAR method. For HAPMS in particular, compensatory/permisive mutations might be expected to be beneficial. However, this does not mean they will be all detected and scored as a PSS. That is because (as discussed) dN/dS estimation methods are rather conservative and can fail to detect all instances of selection. Further some HAPMs may be selectively neutral in the absence of a pCS (LP phenotype), but offer a positive effect when a pCS is present (HP phenotype), and thus may not be inferred as PSS. Regarding BSA and aBSREL, we observed differences in the number of clusters and sites that were significantly scored, but not in the proportions of sites and in different sites or clusters (Supplementary Table 1). Note that BSA and aBSREL were used to detect positively-selected lineages, whilst global ω estimates and PSS were scored under site-specific models. In an attempt to harmonize our findings, the main results reported in Supplementary Table 1 represent a summary of all analyses, in which all lineages and PSS were scored at least under two different methods. In response to the reviewer's point, the following paragraphs have been added in the main text:

To avoid confusion in our methodological approach, we have modified the following sections:

Methods section 5. Testing for diversifying positive selection

Results section "HAPMs associated with diversifying positive selection"

(8) The authors report p-values of 0.0, which does not make sense statistically. These p-values should be reported as less than some value rather than 0.

We agree. We have modified Table 1 accordingly, to show both p-values and significance (above 95% (*), and above the 99.7% (**)) percentiles (Table 1).

Minor revisions

(9) Line 93, the authors write that they confirm high-path virus status by case reports. How as this done for wild bird outbreaks? Additionally, how were sequences classified into domestic and wild bird, especially for duck sequences?

All outbreaks representing the HP clusters selected for this study are reported in the literature (see references in Table 1, main text); these usually include additional information on the host species (see Figure 2). As part of the preliminary analysis stated in **L502-512**, to further assign species (domestic vs. wild bird) for HP outbreaks, we used key terms within headers to relate them to the most commonly farmed avian species (chicken, turkey, duck, quail, poultry, goose, guinea fowl, duck) and classified these as 'domestic'. All headers with scientific names referring to other avian species were tagged as 'wild'. Headers with 'environmental' and 'ND/NA'; were excluded from this analysis. We recognize that our method here is not perfect, and therefore, this data is not shown. It is difficult to ascertain whether duck sequences are from farmed or wild birds; given the overall bias toward poultry sequences, especially in Asia, it is better to assume they are virus sequences from ducks come from farmed animals.

Thus, we further state that "the strength of this association is difficult to assess in nature given the substantial bias (in both the literature and sequence databases) towards AIV infections and HP outbreaks in poultry versus those in wild birds" (**L504-505**).

(10) Lines 125-126, point iii, the wording here confused me a bit. Does this mean that the HAPMs were not observed on a single other internal branch in the whole phylogeny, or that they were not observed on any other internal branch preceding a different high-path cluster? Additionally, this means that convergence could not be detected?

We have attempted to clarify this in the Methods section 3. "Detecting candidate HAPMs co-occurring across HP clusters" (now corresponding to **L167-179**), as follows: "*We identified sites as candidate HAPMs if: (i) they occurred within 2 or more HP clusters, (ii) they were present in >60% of the sequences within the HP clusters (i.e. the mutation is dominant within a given HP population), and (iii) they were absent from internal nodes that do not lead to HP lineages. Thus, candidate HAPMS may occur on LP terminal branches across the phylogenies, but are not generally fixed within LP lineages (Supplementary File 2).*"

(11) The bat polymerase is highly divergent from other influenza A virus polymerases. A discussion of the caveats of modelling amino acid mutations onto this polymerase should be included.

Fortunately, the structures of the H3N2 and H5N1 influenza A viral polymerase have been very recently determined (Ref 38). Thus, we have updated Figure 4 and 5 to use the H5N1 model (PDB: 6QPF), instead of the bat IV polymerase. This improves the accuracy of our analysis. There are no changes to our conclusions. The main text and corresponding figures have been adjusted to account for this.

(12) Figure 2 could be simplified greatly by replacing the maps in panels A and C with smaller tables positioned in the whitespace of the corresponding phylogenies where each table would contain the information shown in tooltips and the pCS

sequences could be shown aligned to each other along with the consensus pCS reported for each region in the text. It was difficult to connect the consensus pCS in the text with those reported in the figure, in the current design.

The purpose of Figure 2 is to represent the spatial and temporal occurrence of the HP clusters for both virus subtypes selected for this study (panels A and C), whilst panels B and D show that the spatiotemporal independence of HP clusters also corresponds to independent genetic lineages. Therefore, deleting the maps (which also give useful background context) would be unhelpful for the general reader. However, we have modified Figure 2 to show the consensus pCS sequences corresponding to each cluster next to the phylogenies. We have also modified panels A and C by removing the pCS sequences and by adding the host species in which the outbreaks occurred according to the literature. We have amended the figure legend correspondingly.

(13) HAPMs shared by high-path clusters were identified visually, but as a sanity check and for reproducibility of the analysis it would be nice if these associations could be algorithmically identified.

We would like to automate the process of identification for HAPMs based on the reconstruction of ancestral states (RAS-ML) under established criteria (stated in **L174-179**), but this would be exceptionally difficult and time-consuming task, taking months of effort for only marginal benefit to other researchers. If the whole analysis we report was to be repeated hundreds of times then it would be worth making this effort, but that's not the case. Our criteria are clearly defined are reproducible; the associations have been cross checked multiple times.

(14) On line 258, a potential control for the method is described based on high-path clusters from the full phylogeny that did not make it into the subsampled phylogeny. This was surprising as the methods for the subsampling suggest that all high-path clusters are included.

We used the large-scale phylogenies to initially identify all HP lineages that that have circulated up to date. However, only a subset of HP clusters that met our criteria were further selected for the analyses. These criteria are defined on **L123-132**: *“Firstly, clusters of HP sequences were further sub-selected under the following criteria: (i) they are monophyletic and well supported (bootstrap value >80) in the large-scale phylogenies of HA and other genome segments, (ii) they contain >2 HP sequences for all genome segments, (iii) they have identifiable immediate LP ancestors, and (iii) they emerged independently from other HP clusters, and are thus distinct in time and space. Eight HP clusters were selected from ten H5NX and fifteen H5N1 previously-identified HP lineages 11 , and nine HP clusters were selected from twenty-four H7NX HP lineages 11. HP sequences were excluded from our analyses if they did not form well-supported clusters, had no identifiable immediate LP ancestors, or if sequences for all genome segments were not available.”*

The occurrence of HAPMs among HP viruses that were not in the datasets represent a natural validation. Thus, we re-estimated the full evolutionary history for a selected number of HAPMs by mapping their amino acid changes onto the tips of the large-scale ML phylogenies generated in step 1 (Supplementary File 3) (**L214-219**). These results indicate that some HAPMs do occur in HP viruses that were not selected, constituting an independent assessment of our statistical approach, and has been now clarified in the main text (**L349-360**).

(15) Statements in the results section could be strengthened by replacing unspecific wording like “several”, “most”, “strong”, or “a variety” with specific counts and/or percentages. Similarly, on lines 39 and 40, are the “over ten” and “over fifteen” pCSs equal to 11 and 16, respectively?

We have made the suggested change.

(16) Lines 140-142 read “Additionally, we re-estimated the full evolutionary history of these HAPMs by mapping the amino acid changes onto the large-scale ML phylogenies generated in step 1.” The meaning of this isn't clear. How were these amino acid changes mapped back onto the ML phylogenies, which in theory may have different topologies, tips, and internal nodes, and why was this performed?

Please see our previous reply for the previous comment (6) for details. ML trees and the time-scaled phylogenies were consistent in their topology, with some differences in branch lengths. This has been now clarified in the main text (**L334**).

(17) In supplemental methods, a typo reads, “ π_T and π_N represents the frequencies of the non-target (T) and target traits (N) states, respectively.” I believe that it should read, “ π_T and π_N represents the frequencies of the target (T) and non-target traits (N) states, respectively.”

Thanks, we have corrected this.

(18) Lines 259 and 281, the “if” should be removed, so they read, “we investigated whether these lineages...” (259) and “Consequently, we investigated whether some HAPMs...” (281).

We have amended as suggested

Reviewers' comments:

Reviewer #1 (Remarks to the Author):

Thank you for considering my comments and for your detailed explanations and modifications to the manuscript, which are all excellent, and I have no further comments to be addressed.

R1.1 [explanation of new statistical model]

Thanks for the clarification this is now OK.

R1.2 [permissive and compensatory mutations]

Super - many thanks for considering this point in depth in your reply, and your modifications in the Methods (L78-92, L167-178, L200-218), Results (L324-559), Discussion (L470-484) sections look excellent.

R1.25 [permissive and compensatory mutations, reassortments]

Modified text L293-296 is good, and this point is also well covered above (R1.2).

R1.3 [explanation of NA data and alignments]

Thanks for your clarification in your response, and the more detailed explanation added in the methods is sufficient.

R1.4 [LP ancestors and other subtypes in the analyses of internal segments]

Excellent, it is good that you were able to consider this point and have modified the data set and re-run the analyses accordingly. The updated results look good.

R1.5, R1.6, R1.7, R1.8, R1.10, R1.16, R1.18, R1.21, R1.24

Thanks for these clarifications

R1.9 [update Figure 1]

Thanks, this figure now looks much clearer.

R1.11 ["all hosts"]

Thanks for this clarification - and I agree that the non-avian other (apart from the Equines) probably don't establish their own lineages, so your methodology seems good.

R1.12 ["all coding"]

Thanks for this clarification - this methodology seems fine.

R1.15 [NA stalk deletions, association with chickens rather than high-path]

Thanks for considering this point, and your additional paragraph (L317-323) provides a reasonable explanation.

R1.17 [ancestral state reconstructions and discrete traits]

Thanks for your explanation

R1.19 [Option of other substitution models] & R.20 [Github link]

That is good news, and I look forward to trying these models.

R1.22 [protein structure modelling]

Updates much appreciated, and pleased that an H5N1 model can now be used (also with no conclusion changes).

R1.23 [clade numbering]

Good, many thanks for including the H5 clade numbers (figures and text) and checking on the H7 scheme.

R1.26, R1.27

The additions to the manuscript look good.

Thanks again for your consideration of my comments, extensive comments and modifications made to the ms.

Reviewer #2 (Remarks to the Author):

Some of my previous technical concerns have been "more or less" included or justified in this new version. However, my major concerns remain. As already indicated in the previous revision, I do not think that this study presents an enough impact for publication in a journal of the level of Nature Communications. Studies like this one are common in journals with a much lower impact.

Results derived from this study are expected, parallel evolution is common in viruses, as already stated by the authors, and association of parallel mutations with pathogenic variants is known and not surprising just because of selection.

This study does not provide any new genetic data. The data was downloaded from a database and analyzed with a battery of traditional phylogenetic estimation programs (phylogenetic tree inference, estimation of selection by dN/dS, etc). A decade ago, this type of pure computational analysis was interesting but nowadays something else (i.e., provide new and relevant data to the scientific community or presenting a highly impacting and unexpected result) is required for publication in highly recognized journals like Nature Communications.

Therefore, my recommendation is reject and suggest transference of the study to a less impacting journal of the Publishing Group.

Reviewer #3 (Remarks to the Author):

The authors devoted significant text responding to comments raised by all 3 reviewers. The authors have added a series of important details into the manuscript and supplement that greatly improve the manuscript. In particular, adding details to clarify how internal and NA gene segments were dealt with, clarifying that human sequences were not included, additional discussion regarding differential sampling of LP and HP viruses in the discussion and supplement, and general addition of information in the Methods is much appreciated. However, there remain aspects of the manuscript that were

difficult to understand or not clearly justified.

Major comments

1. The authors devote significant text in their rebuttal letter to responding to our concerns with population structure and uneven sampling of HP and LP viruses. The authors provide a good explanation for why undersampling of LP viruses may lead to a conservative inference of HAMPs. However, the authors also argue strongly that their method leaves no room for false positives. This overconfidence overlooks one clear concern with undersampling illustrated below. In the example tree below (Tree A), a HP cluster of related viruses is inferred, along with their immediate LP ancestors. In this case, the marked mutation (mut) is inferred as a HAMP. In tree B, more LP viruses have been sampled. When these newly sampled LP viruses are added to the tree, they nest within this HP cluster, breaking it into 2 closely related clusters from distinct origins. The mutation inferred as a HAMP in the tree A is no longer inferred as such tree B. In this case, failing to sample the LP clade introduced in tree B would result in false positive inference of the identified HAMP in tree A.

As explained in our initial review, we recognize that uneven sampling of HP and LP viruses is inherent to avian influenza. We requested, as a control, a subsampling analysis in which the dataset was subsampled, preferably multiple independent times, to generate datasets with 50:50 mixtures of LP and HP viruses. Although the authors performed a subsampling analysis, their procedure was not random, occurring after HP clades were already inferred. To properly account for the uncertainty in HP clusters, this subsampling should be random and occur prior to any analysis. Specifically, the authors should randomly subsampled LP and HP sequences at random from an alignment prior to any tree construction, and perform sensitivity analyses assessing the robustness of their cluster definitions to sampling. This is critical because how HAMPs are defined depends on presence in a cluster. This means that consistent phylogenetic topologies, indeed ones that do not vary depending on sampling, are critical to the accuracy and relevancy of this work. We consider addressing this concern critical to acceptance of the manuscript.

2. The manuscript takes great pains to stress that the statistical model used accurately accounts for sampling bias by conditioning on the initial frequencies of HP and LP sequences. However, there is virtually no main text describing the model or how this model solves the issue of sampling bias. Because the utility of this model is the real heart of this paper, it seems critical that this is addressed in the main text. Furthermore, the authors should devote more care to describing in concrete details how their model overcomes this issue in the main text, rather than relying on the reader to delve into the supplement in great detail.

3. In our initial review, we expressed interest in a greater explanation of the rationale for observing compensatory or permissive mutations in non-HA segments of the genome. The authors, in their rebuttal letter write,

“The flu genome contains so few genes that all must work together in a highly-coordinated fashion and mutations in virus proteins may have pleiotropic effects. Our understanding of these virus protein/protein interactions (direct and indirect) is so incomplete that it’s productive to hypothesise scenarios here. We hope our work will stimulate and direct new experimental work on these questions.”

This explanation is quite vague and does not offer a concrete rationale for why this hypothesis makes sense biologically. The authors should, at minimum, include in the revised manuscript concrete examples of why mutations in non-HA parts of the genome would be hypothesized a priori to impact polybasic cleavage site acquisition. For example, the authors cite PB2 627K and its role in temperature dependence. When a pCS renders a virus enhanced tissue tropism in a chicken, are there specific body

compartments that become infected that are a different temperature than the gut? Furthermore, the authors add this line to line 495-496, "Our findings are also in agreement with previous works proposing that the acquisition of a pCS may require coadaptation of the HA and other viral proteins." The cited reference refers to a single example of 2 M2 variants emerging within-host in an experimentally infected chicken. This is not a robust reference to justify this claim. The authors should find other examples to support their claim in this line, or soften this statement.

Minor comments

1. Lines 147-154 describe the procedure for selecting related internal genes for phylogenetic reconstruction. This was a nice addition in response to reviewer 1, and the effort here is much appreciated. However, as written in the Methods currently, it is quite unclear why this method is being performed. This reviewer spent quite a bit of time reading this section, the rebuttal letter, and the original reviewer 1 comment before understanding why this procedure was employed. The authors should add a sentence or two to this part of the Methods explaining why they are performing this procedure (due to reassortment and the fact that immediate internal ancestors could have been associated with other subtype HAs).

2. In lines 260-263: what does it mean to weigh these scores under a phylogenetic background?

3. We suggested that the authors, as a sanity check, automate their cluster selection. In response to that suggestion, the authors write:

"We would like to automate the process of identification for HAPMS based on the reconstruction of ancestral states (RAS-ML) under established criteria (stated in L174-179), but this would be exceptionally difficult and time-consuming task, taking months of effort for only marginal benefit to other researchers."

It seems to this reviewer that the criteria described on lines 174-179 are not technically difficult and could be implemented. Given the core role of HAPMS to this paper, the ability to accurately re-identify these mutations in a reproducible manner is of immense benefit to other researchers. The authors even mention the importance of re-performing the analyses described in this paper in the future when more sequences are available. Automation of the core HAPMS identification will make this task easier to perform and less error-prone.

6. The portrayal of R_0 in the discussion is confusing and incorrect. Firstly, while R_0 might be higher in high density farming settings, that is because the contact rate between chickens is higher. Secondly, by definition the threshold R_0 to sustain epidemic spread is that R_0 must be 1 or greater. No mutation will reduce this threshold such that an R_0 less than one would result in a sustained outbreak.

Figure concerns

There are several instances in which the authors reference observations in particular figures that are very difficult to find and interpret in the figures themselves. Reformulating these figures remains a critical part of increasing the interpretability of this manuscript.

1. Upon our initial review, we proposed that Figure 2 is somewhat confusing and proposed an alternative solution. The authors responded:

"The purpose of Figure 2 is to represent the spatial and temporal occurrence of the HP clusters for both virus subtypes selected for this study (panels A and C), whilst panels B and D show that the spatiotemporal independence of HP clusters also corresponds to independent genetic lineages. Therefore, deleting the maps (which also give useful background context) would be unhelpful for the

general reader.”

Figure 2 does not effectively communicate information about space or time of HP clusters. The maps in A and C inefficiently represent discrete spatial information (e.g., “Mexico” vs. “Texas”) that any general reader will readily be able to understand while obscuring critical temporal information about the different clusters as small text in tooltips. The trees shown in B and D are not time trees, and therefore contain no information about time. They also do not contain any information about geography. To gain information about space or time, the reader needs cross-reference the cluster label with the above map and read the small boxes of text, which is cumbersome. A timeline annotated by cluster name, consensus sequence, and region aligned with time trees in the bottom panels would be much more effective than the current maps and divergence trees.

Additionally, units should also be added to the scale bars for panels b and d, and legends for colors need to be added. In panels A and C, grey represents a background color, while in B and D it represents LP sequences, which is confusing and currently unclear.

Finally, in lines 277-279, the authors write “The large-scale phylogeny of H5Nx viruses shows a basal clade of H5Nx viruses circulating in wild birds worldwide from the late 1960s to the early 2000s (Supplemental Figure 1, Figure 2).” In neither Supplemental Figure 1 nor Figure 2 is any information provided on the tree with regards to time or bird species. In Figure 2, it is unclear what exactly these host species and times refer to. For example, one of the boxes lists an H5N2 Emu sequence from Texas. Were sequences from this cluster entirely derived from Texas Emus in the year listed? It is also unclear what the text regarding time is conveying in the map panels. Because only a single date is shown, are these years the only years in which samples from this cluster were sampled? How long-lived were these clusters? Because the trees also lack time information, there is no way to gain this information from the current figure.

2. The time-resolved phylogenies shown in Figure 3 should be shown with an x-axis that represents time. Additionally, designating HP and LP clusters by orange and grey boxes is difficult to distinguish. Perhaps recoloring the trees or finding an alternative method to indicate HP vs. LP would be helpful.

3. In the Figure 2 legend it says “high-supported clades” are shown in red. How was “high-supported” decided?

4. The legend for Table 1 still lists a p-value of 0.00. This does not make statistical sense and should be replaced.

Typos/grammatical concerns

Line 29: “Virulence influenza A viruses is...” should read “Virulence in influenza A viruses is ...”

Line 340: typo “Evolution at site154”

We thank the reviewer for recognizing our efforts to address all issues raised. We now address the remaining points below.

Reviewer # 3 (Remarks to the Author):

Major comments

(1) The authors devote significant text in their rebuttal letter to responding to our concerns with population structure and uneven sampling of HP and LP viruses. The authors provide a good explanation for why undersampling of LP viruses may lead to a conservative inference of HAMPs. However, the authors also argue strongly that their method leaves no room for false positives. This overconfidence overlooks one clear concern with undersampling illustrated below. In the example tree below (Tree A), a HP cluster of related viruses is inferred, along with their immediate LP ancestors. In this case, the marked mutation (mut) is inferred as a HAMP. In tree B, more LP viruses have been sampled. When these newly sampled LP viruses are added to the tree, they nest within this HP cluster, breaking it into 2 closely related clusters from distinct origins. The mutation inferred as a HAMP in the tree A is no longer inferred as such tree B. In this case, failing to sample the LP clade introduced in tree B would result in false positive inference of the identified HAMP in tree A.

Reply:

Mutation X is only scored as a HAPM if its distribution in the tree coincides with the HP/LP trait, which is assessed using our phylogenetically-informed model. That is, X is only scored as a HAPM if it is found more often among HP lineages vs LP lineages (alternate hypothesis, H1) than expected by chance, i.e. when HAPMs and HP/LP states change interpendently of each other (null hypothesis, H0). Thus, testing of H1 is independent of the observed clustering pattern of HP or LP sequences. The use of the tree structure to inform the model is to correct for the phylogenetic correlations that may bias the test (for example, the effect of the founder effect giving rise to the increase in distribution of a given mutation within a specific lineage). Phylogenetic testing of correlated character evolution is a classic technique in evolutionary biology and we have simply used here a specific extension of that approach (see Harvey & Pagel "The Comparative Method in Evolutionary Biology" OUP, 1991).

Reply:

*In tree B, if a given HP lineage was broken into two distinct clusters with different LP origins, this would NOT lead to a false positive. **That is because either (i) we fail to observe an additional LP to HP change (i.e. our results are conservative), or (ii) the new LP sequences have mutation X, thereby weakening the statistical support for H1.** As an example of this is the H5N1 cluster C8, that splits into two clusters (C8.1. an C8.2) in the PB2, PA and NP trees compared to the HA tree (L308). Yet, this does not have an impact on the statistical support of the H5NX HAPMs detected both in HA and PB2 (see Table 1). All of this is completely theoretical, however, because HP clusters arise rarely, at different points in time and space, and on different virus lineages, it is incredibly unlikely that two HP clusters will be mistakenly aggregated. There is no reason to suppose that has occurred and is it not considered as a problem by the global AIV surveillance community. It is important to stress that the lineages labelled as HP are real, defined by a pCS and shown in multiple other studies. Comparable cluster definitions have been estimated from very different and independent subsampling (see references 5, 11, 28, 40 and 67).*

As explained in our initial review, we recognize that uneven sampling of HP and LP viruses is inherent to avian influenza. We requested, as a control, a subsampling analysis in which the dataset was subsampled, preferably multiple independent times, to generate datasets with 50:50 mixtures of LP and HP viruses. Although the authors performed a subsampling analysis, their procedure was not random, occurring after HP clades were already inferred. To properly account for the uncertainty in HP clusters, this subsampling should be random and occur prior to any analysis. Specifically, the authors should randomly subsampled LP and HP sequences at random from an alignment prior to any tree construction, and perform sensitivity analyses assessing the robustness of their cluster definitions to sampling. This is critical because how HAPMs are defined depends on presence in a cluster. This means that consistent phylogenetic topologies, indeed ones that do not vary depending on sampling, are critical to the accuracy and relevancy of this work. We consider addressing this concern critical to acceptance of the manuscript.

Reply:

This is conceptually linked to our response above. Phylogenetically-informed statistical tests of correlated trait evolution are distinct from standard association tests (e.g. chi-squared test) and do NOT follow the normal rules of statistical sampling. Specifically, the sequences (and their amino acids) are not independent data points, and instead can be highly correlated. Because of this correlational structure, sub-sampling to 50:50 sequence mixture does not do what the reviewer thinks it does: such subsampling is not the right way to evaluate the uncertainty in HP clusters. The independent observations on which our test relies are, instead, the changes in state. Uncertainty in these state changes is correctly addressed in our null simulations, in which the numbers of HP/LP taxa are constrained to the empirical observed ratio. Based on our experience in phylogenetic hypothesis testing, we are confident that our approach is statistically correct.

As explained above, our HP cluster definitions are not a cause of bias here. Inspection of the huge literature on HP AIV surveillance will show the ubiquitous use of phylogenetic character mapping to delineate HP clusters in AIVs. Furthermore, the phylogenetic distribution of HP clusters is conserved both within the large-scale dataset and the subsampled dataset, confirming phylogenetic consistency even if the proportion of LP to HP viruses and number of sequences change. Thus, we have "properly account[ed] for the uncertainty in HP clusters". We hope that we have been able to clarify the conceptual background of our work.

(2) The manuscript takes great pains to stress that the statistical model used accurately accounts for sampling bias by conditioning on the initial frequencies of HP and LP sequences. However, there is virtually no main text describing the model or how this model solves the issue of sampling bias. Because the utility of this model is the real heart of this paper, it seems critical that this is addressed in the main text. Furthermore, the authors should devote more care to describing in concrete details how their model overcomes this issue in the main text, rather than relying on the reader to delve into the supplement in great detail.

Reply:

Unfortunately, there is insufficient room in the main text to derivation and validation of this statistical model. However, this information is available in full in the supplementary material (Supplementary Information: Supplementary Text 5. Full mathematical description of the model and simulations), frequently referred to by the main text. It's not correct that we don't summarise this model and the issue of sampling in the main text. Specifically, our last version included in the main text the following (L489-498): "Available genomes for AIV are likely not randomly sampled. The LP ancestors of many HP clusters are likely under sampled, and some HP lineages may not be sampled at all. However, due to the nature of our phylogenetic tests, these sampling biases will lower our statistical power to detect HAPMs, but they will not introduce false positives. Therefore, our statistical approach is conservative (some associations will not be detected), but robust (the associations detected are real). Further, our simulations already account for potential sampling bias, as the number of HP and LP states were fixed to those observed in the real data, whilst amino acid evolution was simulated under the model conditioned on the fixed trait patterns. As we use a probabilistic approach, we expect our model will only gain power as sampling of HP/LP viruses improves in the future."

Nonetheless, in order to clarify and provide further support for this point we have rewritten and expanded the main text as follows (L490-503): "However, due to the phylogenetic nature of our tests, these sampling biases will lower our statistical power to detect HAPMs, but they will not introduce false positives. This occurs because the test is based on trait state changes on internal phylogeny branches, not on the trait frequencies observed at the tree tips (see 20). Therefore, our statistical approach is conservative (some associations will not be detected), but robust (the associations detected are real). Further, the simulations used to evaluate the performance of our statistical test take account for potential sampling bias, as the number of HP and LP states were fixed to those observed in the empirical data, whilst amino acid evolution was simulated under the model conditioned on the fixed trait patterns (see Supplementary Text 5 for details). Reconstructed trait state changes were consistent between the large-scale and the subsampled trees, further confirming that our model is insensitive to substantial fluctuations in the frequency of sampling HP/LP sequences. As we use a probabilistic approach, we expect our model will gain power as sampling of HP/LP viruses improves in the future."

(3) In our initial review, we expressed interest in a greater explanation of the rationale for observing compensatory or permissive mutations in non-HA segments of the genome. The authors, in their rebuttal letter write,

"The flu genome contains so few genes that all must work together in a highly-coordinated fashion and mutations in virus proteins may have pleiotropic effects. Our understanding of these virus protein/protein interactions (direct and indirect) is so incomplete that it's productive to hypothesise scenarios here. We hope our work will stimulate and direct new experimental work on these questions." This explanation is quite vague and does not offer a concrete rationale for why this hypothesis makes sense biologically. The authors should, at minimum, include in the revised manuscript concrete examples of why mutations in non-HA parts of the genome would be hypothesized a priori to impact polybasic cleavage site acquisition. For example, the authors cite PB2 627K and its role in temperature dependence. When a pCS renders a virus enhanced tissue tropism in a chicken, are there specific body compartments that become infected that are a different temperature than the gut? Furthermore, the authors add this line to line 495-496, "Our findings are also in agreement with previous works proposing that the acquisition of a pCS may require coadaptation of the HA and other viral proteins." The cited reference refers to a single example of 2 M2 variants emerging within-host in an experimentally infected chicken. This is not a robust reference to justify this claim. The authors should find other examples to support their claim in this line, or soften this statement.

Reply:

As suggested, we have softened this claim and added some additional support within the discussion section (L457-468) as follows:

"At present we can only hypothesise on the mechanism underlying these associations and whether they if they might have arisen due to epistasis, which is thought to be common in RNA viruses due to their limited genome sizes (14). As an example of within-gene epistasis, oseltamivir resistance in human H1N1 influenza A viruses is conferred by mutation H274Y in N1, yet appears to have spread only under the genetic background of two permissive mutations (V234M and R222Q also in N1), that enable the virus to tolerate the detrimental effects of H274Y on protein structure stability (Ref 18). Further, PB2 mutations E627K and A674T both scored as HAPMs, appear to epistatically interact: A674T counteracts the structural stability induced by E627K (Ref 63), whilst in vivo mouse models have shown that this mutation is a prerequisite for the acquisition of E627K, by stabilizing polymerase activity (Ref 64). Epistatic interactions among IAV genes are much less studied but have been proposed for genes of the IAV polymerase complex (63, 64)."

We again stress that we do not need to hypothesise a functional mechanism for the associations we observe in order for the associations to be real. We can statistically reject the hypothesis that the associations arose by chance. As evolutionary biologists, we must leave the task of understanding the mechanism behind these associations to molecular biologists.

Minor comments:

1. Lines 147-154 describe the procedure for selecting related internal genes for phylogenetic reconstruction. This was a nice addition in response to reviewer 1, and the effort here is much appreciated. However, as written in the Methods currently, it is quite unclear why this method is being performed. This reviewer spent quite a bit of time reading this section, the rebuttal letter, and the original reviewer 1 comment before understanding why this procedure was employed. The authors should add a sentence or two to this part of the Methods explaining why they are performing this procedure (due to reassortment and the fact that immediate internal ancestors could have been associated with another subtype HAs).

Reply:

We thank the reviewer for this feedback. We now clarify the rationale for the analysis in Methods section (L151-157) as follows: "AIVs of different subtypes undergo genome reassortment, especially within the internal segments"²⁴. In order to account for this, when inferring the evolutionary history of internal genes, we used a single phylogeny/dataset that includes all AIV subtypes (not just viruses belonging to the H5 and H7 subtypes). In this way, we maximise our chances of including phylogenetic nodes that are closely-related LP ancestors to HP clusters, irrespective of AIV subtype. This procedure aims to (i) reduce estimation error, (ii) increase accuracy of phylogeny reconstruction, and (iii) ensure the most closely related strains to each HP lineage are included."

2. In lines 260-263: what does it mean to weigh these scores under a phylogenetic background?

Reply:

We have now clarified the sentence as follows (L272-275): "A statistical model is used to calculate the probability of free energy change along phylogeny branches in a manner analogous to the computation of nucleotide substitution models, and thereby computationally predict the occurrence of mutations altering protein stability (Ref 37)."

3. We suggested that the authors, as a sanity check, automate their cluster selection. In response to that suggestion, the authors write: "We would like to automate the process of identification for HAPMS based on the reconstruction of ancestral states (RAS-ML) under established criteria (stated in L174-179), but this would be exceptionally difficult and time-consuming task, taking months of effort for only marginal benefit to other researchers.". It seems to this reviewer that the criteria described on lines 174-179 are not technically difficult and could be implemented. Given the core role of HAPMS to this paper, the ability to accurately re-identify these mutations in a reproducible manner is of immense benefit to other researchers. The authors even mention the importance of re-performing the analyses described in this paper in the future when more sequences are available. Automation of the core HAPMS identification will make this task easier to perform and less error-prone.

Reply:

Given current circumstances it is not possible to automate our pipeline. Further, we stress that this is not a trivial task, sufficient to warrant a publication of its own, yet this is within our plans, as we agree that the availability of this pipeline will add value to the research community. Nonetheless, we are confident that our careful data collation is correct, and that an automated process will have no effect on our results or conclusions. Moreover, automated approaches are not guaranteed to be error-free and need to be carefully revised. Nonetheless, this may take months, and due to the loss of key computational staff (to industry, and seconded to COVID work), we will not be able to complete this additional task in the near future.

4. The portrayal of R0 in the discussion is confusing and incorrect. Firstly, while R0 might be higher in high density farming settings, that is because the contact rate between chickens is higher. Secondly, by definition the threshold R0 to sustain epidemic spread is that R0 must be 1 or greater. No mutation will reduce this threshold such that an R0 less than one would result in a sustained outbreak.

Reply:

The reviewer is correct, we incorrectly stated "threshold R0" instead of "threshold density". Since $R_0 = \beta \cdot c \cdot D$, where c is contact rate, it is correct to say that higher contact rates lead to higher R0. Note that R0 is a property of an epidemic, not of a pathogen, and therefore depends on both pathogen and host factors.

These corrections have now been made as follows (L550-558): "This suggests that the parallel mutations identified here are not directly linked to transmission among domestic bird populations, although high population densities together with a limited genetic variability within the host (as occurs in intense farming), may facilitate virus transmission. If acquiring a pCS does indeed reduce AIV transmission fitness in natural systems 50, then it is interesting to speculate that both intensive farming and the occurrence of secondary mutations (HAPMs) may independently act to facilitate the spread of a HP strain; the latter by potentially increasing the transmission fitness of strains carrying a pCS, and the former by lowering the threshold host density required to sustain transmission in a population."

5. Figure 2 does not effectively communicate information about space or time of HP clusters. The maps in A and C inefficiently represent discrete spatial information (e.g., "Mexico" vs. "Texas") that any general reader will readily be able to understand while obscuring critical temporal information about the different clusters as small text in tooltips. The trees shown in B and D are not time trees, and therefore contain no information about time. They also do not contain any information about geography. To gain information about space or time, the reader needs cross-reference the cluster label with the above map and read the small boxes of text, which is cumbersome. A timeline annotated by cluster name, consensus sequence, and region aligned with time trees in the bottom panels would be much more effective than the current maps and divergence trees. Additionally, units should also be added to the scale bars for panels b and d, and legends for colors need to be added. In panels A and C, grey represents a background color, while in B and D it represents LP sequences, which is confusing and currently unclear. Finally, in lines 277-279, the authors write "The large-scale phylogeny of H5NX viruses shows a basal clade of H5NX viruses circulating in wild birds worldwide from the late 1960s to the early 2000s (Supplemental Figure 1, Figure 2)." In neither Supplemental Figure 1 nor Figure 2 is any

information provided on the tree with regards to time or bird species. In Figure 2, it is unclear what exactly these host species and times refer to. For example, one of the boxes lists an H5N2 Emu sequence from Texas. Were sequences from this cluster entirely derived from Texas Emus in the year listed? It is also unclear what the text regarding time is conveying in the map panels. Because only a single date is shown, are these years the only years in which samples from this cluster were sampled? How long-lived were these clusters? Because the trees also lack time information, there is no way to gain this information from the current figure.

Reply:

Clearly, the purpose of our study was not to produce a systematic review of the dates and locations of AIV HP outbreaks worldwide. There are many other data sources that achieve that. For example, HP genotypes have been extensively detailed in Ref 11 (Official OFFLU document http://www.offlu.net/fileadmin/home/en/resource-centre/pdf/Influenza_A_Cleavage_Sites.pdf citing over 45 different papers). The metadata with all information associated with the HP viruses used in this work (both the H5N2 and H7N2 subtypes) is available in Supplementary File 1, Accessions (HPAIV sheets).

Instead, Figure 2 aims to provide a succinct summary of the known information about HP clusters that is relevant to our particular study. Figure 3 displays the phylogenetic position of HP clusters, and therefore there is no need for them to contain temporal or geographic information. Regarding lines 277-279 (now L312-316), we again stress that the HP phenotype/genotype definition we used is based on factual reports of HP outbreaks (see references 5, 11, 28, 40 and 67). Our phylogenetic analysis simply confirms that these outbreaks correspond to independent genotypes (lineages) with different pCSs (Figure 2 panels B and D, and Supplemental Figure 1).

HP outbreaks tend to be detected within single farming facilities that handle one or just a few avian species, hence typically all sequences within a single cluster belong to the same species. In Panels A and C, when a single avian species has been reported for a given outbreak, this is stated as such in the corresponding box (example H5N2 C1, Emu). On the other hand, when several avian species were reported, this is also stated as such (example H7N1 C2, mixed poultry). Similar details are given for the geographical locations and dates shown. However, sometimes the documentation for an outbreak is not complete or has not been published, it is not possible to retrieve all missing information (example, H5N2 C1). Other well-documented outbreaks (e.g. H7N1 C2 HP outbreak in Italy) occurred in multiple farmed species lasting a whole year (1999-2000) (see Refs 66-68), and this is also stated as "mixed". Thus, given the limitations of surveillance efforts, it is impossible to know the exact distribution within hosts, geographic regions and duration for every outbreak, especially those further back in time.

In response to the reviewer's feedback we have made the following changes:

"Additionally, units should also be added to the scale bars for panels b and d"

- This has been added (substitutions/site), as suggested. Text size for the boxes has also been enlarged.

"Legends for colors need to be added"

- This is now fully described in the corresponding figure legend.

"In panels A and C, grey represents a background color, while in B and D it represents LP sequences, which is confusing and currently unclear"

- This has now been modified by changing colouring in both trees, using black to represent LP branches

We have added four new Supplementary Figures (5-8) based on Figure 3, that show the trees in Figure 2b and 2d in a larger size and with full taxon labels, which includes information on sampling date, location, and host species.

6. The time-resolved phylogenies shown in Figure 3 should be shown with an x-axis that represents time. Additionally, designating HP and LP clusters by orange and grey boxes is difficult to distinguish. Perhaps recoloring the trees or finding an alternative method to indicate HP vs. LP would be helpful.

Reply:

Please read our response to point raised above, which explains why a time-axis isn't necessary in Figure 3. However, in order to help readers, we now provide a larger and more detailed versions of all trees in Figure 3 in the Supplementary Information (Supplementary Figures 5-8). These trees now include taxon names, make it easier to discriminate the HP and LP clusters, and they also include a time scale.

To avoid further confusion, we have also modified the title of the Figure 3 from "Time-scaled reconstruction of amino acid evolution at selected HAPMs" to "Reconstruction of amino acid evolution at selected HAPMs" (L670-671).

7. In the Figure 2 legend it says "high-supported clades" are shown in red. How was "high-supported" decided?

Reply:

The definition of well-supported clusters is indicated in the methods section (L126-128): "Firstly, clusters of HP sequences were further sub-selected under the following criteria: (i) they are monophyletic and well supported (bootstrap value >80) in the large-scale phylogenies of HA and other genome segments...". We have also modified the sentence in the figure legend as follows (L635-637): "well-supported lineages (as defined in methods, indicated by C1-C9, showing posterior probability and bootstrap support values >80 or 0.8 for nodes of interest)."

8. This does not make statistical sense and should be replaced.

Reply:

*This was simply a rounding issue. We have reformatted the numbers in Table 1 to two significant digits. To avoid further confusion, column 9 in Table 1 now shows significance using a cut-off value of <0.05 or of <0.01 using the symbols * or with **, respectively.*

Typos/grammatical concerns

Line 29: "Virulence influenza A viruses is..." should read "Virulence in influenza A viruses is ..."

Line 340: typo "Evolution at site154"

Reply:

Both typos have been corrected.

Reviewers' Comments:

Reviewer #1:

Remarks to the Author:

Although I was happy with the author's previous revisions - the authors also made some changes in response to one of the other reviewers. The other reviewer raised some particular technical issues which the authors have also responded to. So my comments and discussion below are mostly on these technical issues.

(small point 1) from Reviewer 3

"The additional supplemental figures are great. They are significantly easier to read and understand. I believe that it would improve readers' ability to understand the authors conclusions if the trees with this level of detail were part of the main text. Perhaps the authors could show one of these new trees instead of the 4 in current figure 3."

My comments - I also like the trees in the supplementary with the dual coloured bars; and the reviewers suggestion is reasonable, but the trees as they are in the main text are OK too.

Technical Point 2 from Reviewer 3 on Subsampling

"We acknowledge that the subsampling regime we suggested may not be necessary. It is fine to proceed without, and we appreciate the explanation."

In the paper the subsampling method is in lines 122-146.

My comments are - the authors subsample the data in a stratified and non-random way. This method of subsampling is suitable for phylogenetic studies because it retains the important HP clusters and related LP sequences.

One of the problems with other more 'black box' statistical methods is that they fail to take the underlying (viral) population structure into account properly and thus end up with either biased or essentially meaningless conclusions e.g. this mutation is associated with that phenotype; but it is actually just a founder effect that has not been accounted for.

So yes, I think the subsampling method the authors used is good and appropriate (and is in fact preferable to the random subsampling that reviewer 3 suggests), and I also think that the authors response to this point (in round 2) is valid.

Technical Point 3 from Reviewer 3 on false positives

This is a complicated issue, and I think reviewer 3 is right - in that there is the technical possibility of false positives in the manner they describe. And I think that it is related to apparent HP -> LP transitions.

Roughly "if some newly sampled LP was within an previously thought of HP monophyletic cluster, and the LP was missed in the subsampling procedure then there would be a false positive.

However, I don't think this case happens in nature (or in transmission experiments - it is usually the other way around and possibly related to vaccine experiments which do not do the same thing as wild strain dynamics anyway.

My understanding is that there are LP to HP transitions in nature, and on the tree, but not the other way around, i.e. there are no cases of a previously highly pathogenic Hemagglutinin (with the multi-basic cleave site) ever losing that genome region. Sure the sub-sub-clade might die out in a spatial location, and then a new LP strain come in from elsewhere, but that is not the same thing as the sub-sub-clade actually reverting to LP. So would a false positive be the correlation of a mutation at a different site to a false LP -> HP transition, which would be a HP -> LP transition ? But these have never been observed in nature.

Is there a possibility that the LP/HP reconstruction on an HA tree could occasionally show a false HP -> LP transition then ? Well yes perhaps - especially if a symmetric binary trait model was used (as opposed to an asymmetric one), and/or the tree was poorly resolved, this is technically a possibility. But, even if it may be technically possible, considering the real structure of the trees and the genetic distance of the HP clusters, then I don't think that this technical error is happening in these data.

Is there a possibility that a LP/HP binary trait reconstruction on a non-HA tree could show a HP -> LP transition ? yes, this could happen, it could be that from the point of view of a non-HA segment, it could acquire and lose its HP HA partner through reassortment. And looking at the trees in the supplementary, it does look like there are apparent HP -> LP transitions on internal branches. But are these false transitions ? Looking at top part of NP of H7NX in Supplementary figure 2, we see C7 (H5N2) -> C8.1 (H5N1), then either C7 -> LP or C8.1 -> LP, then LP -> C1 (H7N9). So this might be a place where a false positive could occur. The split C8 group in internal segments is also referred to in lines 307-309.

But having said all of this, what the authors have actually claimed in the manuscript is OK though (e.g. lines 67-69, 72-74, 178-185, 480-484, 514-517). And lines 480-482 seem OK as they are (in context of sampling biases) - but could be shortened to be less contentious if wanted (just keep the first part of the sentence remove the second so that it becomes):

'However, due to the phylogenetic nature of our tests, these sampling biases will lower our statistical power to detect HAPMs'

And maybe remove '(the associations detected are real)' in line 484/485

Also the simulation results (in the supplementary) already show the point about sampling bias and statistical power.

Technical point 3b of Reviewer 3 - biologically plausible associations

Firstly to test in real experiments whether pairs of mutations enhance or suppress virulence or adaptation properties of avian influenza viruses in birds is a large and complex experimental undertaking, and as we are talking about highly pathogenic avian influenza, can only be done in a few facilities in the entire world, and would be subject to gain-of-function ethical considerations. Consequently, although there are some studies which try to address these, it is very much an area where guidance from statistical and phylogenetic studies of wild strains in nature can be very useful in helping to choose which pairs of mutants to try to test various hypotheses.

So whilst I agree with the reviewer that it would be good to say if there is any corresponding experimental evidence for the associations found, and to suggest why there might be an association detected, this information does not necessarily exist in the literature. The authors do point out that the existence of associations are only a hypothesis anyway in lines 53-61 of the introductions, there is

a section where the authors discuss possible functional relevance of mutations (lines 404-422), and the authors do discuss some associations in the discussion section (lines 448-459).

Minor point - line 507 - maybe just add why you cant really tell if permissive or compensatory even with a time scaled tree ? e.g. something like this ? —

However, we were not able to infer with sufficient resolution the relative timing of occurrence of most HAPMs under our time-scaled phylogenies, in part due to the longer branches leading to some of the HP clusters.

Minor point 5 of Reviewer 3 - about Figure 2 and occurrence of HP worldwide & author's response
"Clearly, the purpose of our study was not to produce a systematic review of the dates and locations of AIV HP outbreaks worldwide. There are many other data sources that achieve that. For example, HP genotypes have been extensively detailed in Ref 11"

Figure 2 seems fine to me because it shows when and where the marked groups (C's) are for H7 and H5. And ref 11 is good to include.

Reviewer #3:

Remarks to the Author:

1. The additional supplemental figures are great. They are significantly easier to read and understand. I believe that it would improve readers' ability to understand the authors conclusions if the trees with this level of detail were part of the main text. Perhaps the authors could show one of these new trees instead of the 4 in current figure 3.

2. We acknowledge that the subsampling regime we suggested may not be necessary. It is fine to proceed without, and we appreciate the explanation.

3. We remain unconvinced that the authors have fully dealt with issues of false positives. The authors essentially extend published methods of Pagel et al, which have been widely used for trait associations. However, as discussed in this article (<https://academic.oup.com/sysbio/article/64/1/127/2847997>), these methods can result in spurious positive associations under a couple of common phylogenetic scenarios. We lack the expertise to fully evaluate whether the simulations provided by the authors adequately show that their new method is immune from these concerns, and the authors do not provide justification for their claims or extensive rationale for why the concerns highlighted by Madison et al do not apply to avian influenza phylogenies. This is particularly problematic because of the authors' insistence that it is not their job to assess whether the associations they identify are biologically plausible. As noted by others, consideration of whether correlations are biologically logical is a critical component of these types of analyses. In response to our request to provide greater explanation for the rationale that pCS acquisition would be accompanied by compensatory mutations in non-HA segments, the authors note that "We again stress that we do not need to hypothesise a functional mechanism for the associations we observe in order for the associations to be real. We can statistically reject the hypothesis that the associations arose by chance. As evolutionary biologists, we must leave the task of understanding the mechanism behind these associations to molecular biologists." This stance is problematic given that the authors also advocate for their identified sites being incorporated into avian influenza surveillance practices in the final line of the manuscript.

Given these concerns, we recommend the following: if the model described in this paper truly ameliorates long held issues with phylogenetic trait correlations, this is a major finding that should be highlighted extensively in the main text. If not, then the authors need to more fully acknowledge the limitations of their model. Because phylogenetic trait correlations is an entire subfield, we do not have

the expertise to fully evaluate whether this is the case. We recommend that the manuscript be reviewed by an additional reviewer with the expertise to evaluate the model, specifically a statistical phylogeneticist.

Final revisions for Nature Communications manuscript NCOMMS-19-27454B

RESPONSE TO REVIEWERS

We are happy to know that our manuscript has been considered for publication in *Nature Communications*. We appreciate the positive feedback and acknowledgement of our efforts in the review process of this manuscript. Following the suggestions made by the reviewers, please see below our final responses and remarks regarding the modifications made to the main text and other manuscript files.

Reviewer #1 (Remarks to the Author):

Technical Point 1: "The additional supplemental figures are great. They are significantly easier to read and understand. I believe that it would improve readers' ability to understand the authors conclusions if the trees with this level of detail were part of the main text. Perhaps the authors could show one of these new trees instead of the 4 in current figure 3." My comments - I also like the trees in the supplementary with the dual coloured bars; and the reviewer's suggestion is reasonable, but the trees as they are in the main text are OK too.

Reply:

As suggested we have now modified the Supplementary Figs 2 and 3 in the Supplementary Information with dual coloured bars on the trees. The following text was also added to the corresponding figure legends: *"Dual coloured bars represent direct ancestry, in which HP clusters are highlighted in orange, whilst the possible immediate ancestral LP sequences are highlighted in grey."* We haven't replaced main text Figure 3 with these, because they are showing different things, and it is important to include both representations. Specifically, supplementary Figs 2 and 3 are ML trees denoting direct ancestry and are not ancestral reconstructions for selected sites. Reviewer 1 states they are happy if we retain the current main text Figure 3.

Technical Point 3: on false positives. Lines 480-482 seem OK are they are (in context of sampling biases) - but could be shortened to be less contentious if wanted (just keep the first part of the sentence remove the second so that it becomes): 'However, due to the phylogenetic nature of our tests, these sampling biases will lower our statistical power to detect HAPMs'. And maybe remove '(the associations detected are real)' in line 484/485. Also, the simulation results (in the supplementary) already show the point about sampling bias and statistical power.

Thanks for the suggestion. We have modified the paragraph in L480-486 as follows: *"Available genomes for AIV are likely not randomly sampled. The LP ancestors of many HP clusters are likely under sampled, and some HP lineages may not be sampled at all. However, due to the phylogenetic nature of our tests, these sampling biases will lower our statistical power to detect HAPMs, but they will not introduce false positives. Therefore, our statistical approach is conservative (some associations will not be detected), but robust. Further, the simulations used to evaluate the performance of our statistical test take account for potential sampling bias (see Supplementary Text 5 for details). As we use a probabilistic approach, we expect our model will gain power as sampling of HP/LP viruses improves in the future."*

Technical point 3b of Reviewer 3 - biologically plausible associations. Minor point - line 507 - maybe just add why you can't really tell if permissive or compensatory even with a time scaled tree? e.g. something like this? However, we were not able to infer with sufficient resolution the relative timing of occurrence of most HAPMs under

our time-scaled phylogenies, in part due to the longer branches leading to some of the HP clusters.

Lines 499-4501 were modified as suggested: *“However, we were not able to infer with sufficient resolution the relative timing of occurrence of most HAPMs under our time-scaled phylogenies, in part due to the longer branches leading to some of the HP clusters.”*